# Virtual reality interactions via a user-generic ultrasound human-machine interface for wrist and hand tracking

Bruno Grandi Sgambato[1], Bálint K. Hodossy[1], Deren Yusuf Barsakcioglu[1], Xingchen Yang [1,2], Anette Jakob [3], Marc Fournelle[3], Meng-Xing Tang [1] & Dario Farina [1] ✉

As computers move from desktop screens into our glasses, traditional controllers such as keyboards and mice have proven impractical. A control interface for immersive experiences needs to seamlessly transport intention from the real to the virtual world while remaining portable, accurate, and robust. Here, we present an easy-to-wear, dry-contact and portable ultrasound armband that can decode morphological information and act as a virtual reality controller by predicting hand and wrist kinematics. Using our armband, we collected a large dataset of paired ultrasound and hand kinematics and used it to train supervised deep-learning models capable of predicting hand kinematics from ultrasound. We explored how diverse intra-session, cross-session, and cross-participant data shifts affect model performance. Further, we proposed methods for data conditioning, augmentation, and a referencing strategy to mitigate the influence of confounding factors and to achieve accurate prediction of hand kinematics on unseen users without fine-tuning. Finally, we demonstrated the feasibility of our interface in a real-time virtual reality control framework. Using the developed ultrasound interface, participants completed challenging interaction tasks with simulated contact physics. This work demonstrates the potential of ultrasound-based technologies as a virtual reality interface, showcasing strong performance, robustness, and generalization potential.

Extended reality (XR) is an umbrella term that refers to the combined fields of virtual reality (VR), augmented reality (AR), and mixed reality (MR). These technologies aim at merging the physical and virtual worlds into one immersive experience. In practice, interleaving virtual and real objects involves a variety of human-machine interfaces (HMI) to both convey user's input to virtual objects and to provide feedback from virtual interactions to the user. Therefore, as HMIs mediate all aspects of user experience, their performance is paramount to the quality of the experience and widespread adoption of XR devices.

Since hands are primary effectors through which humans can interact with the environment, a crucial subset of HMIs focuses on transmitting upper limb positions and motion intentions from the physical to the virtual world. Ideally, these systems need to perfectly transfer real-world movement to a virtual twin that can then interact with virtual objects[1]. Currently, these HMI systems mainly use optical systems, sensorized gloves, and wrist and arm bracelets[2].

Vision-based approaches are the industry standard, being present in multiple commercial solutions (e.g., Meta Quest 3, Leap Motion,

[1]Department of Bioengineering, Imperial College London, London, UK. [2]School of Automation, Southeast University, Nanjing, China. [3]Ultrasound Department, Fraunhofer Institut für Biomedizinische Technik, Sulzbach, Germany. ✉e-mail: d.farina@imperial.ac.uk

Vicon Suite). These systems are able to track multiple upper limb degrees of freedom (DoFs) with high accuracy either via reflective markers or computer vision[3]. However, they are inherently limited by object occlusion, heavily limiting their applicability in complex environments and when interacting with objects. Vision systems also suffer from high cost, and heavy computational and power requirements. When translated to body-mounted cameras instead of fixed ones placed around a capture space, limitations in camera count and field of view lead to a substantial decrease in performance and robustness.

Sensorized gloves, as the name implies, are commonly fitted with a variety of sensors[4] such as IMUs[5], strain[6], or capacitive[7] sensors. When compared to vision systems, they are advantageous due to their simplicity, translating to low complexity and wearability. Sensorized gloves, however, suffer from unstable performance that can hinder manipulation[8] and accumulate sensor drift[9].

Surface Electromyography (sEMG) sensors are the main technology on the closely related field of gesture recognition. By recording the electrical signals generated by muscles during contractions, sEMG systems can leverage complex machine learning (ML) algorithms to map muscle contraction to gesture intent[10,11]. Recent efforts in collecting massive datasets have been boosting system classification performance and generalization[12,13]. However, when used for motion capturing, challenges in mapping sEMG to robust kinematics predictions are still relevant[2]. Challenges for robust tracking include low penetration depth and spatial resolution[11], muscle fatigue[14], and signal drift[15,16], all of which constrain its applicability. In this context, other technologies able to overcome the inherent limitations of current systems remain to be explored.

Ultrasound (US) has been gathering increasing attention as an alternative signal source for HMIs[17,18]. When applied to the upper limb, US can provide cross-sectional or longitudinal morphological information of a user's limb. Its main working principle is the ability to extract shape, size, and location of diverse arm structures (muscles, bones, connective tissue, etc). By tracking these structures over time, a precise mapping to the joint angles can be derived. In simple scenarios, it has been shown that specific morphologically significant structures can be tracked and mapped to arm kinematics. For example, Zheng et al.[19] demonstrated that with the arm lying flat on a table, the deformation of the extensor carpi ulnaris muscle, recorded with an US brightness-mode (B-mode) imaging probe, could be readily correlated with the wrist extension angle. Other studies have expanded this concept to force tracking[20] or implemented similar muscular boundary tracking with a simpler amplitude-mode (A-mode) setup[21]. Nevertheless, the explicit morphological tracking approach is challenging to translate to more realistic scenarios as muscular deformation during unconstrained movements is complex. Precisely identifying and tracking meaningful morphological structures—that often disappear when leaving the imaging plane—is impractical[22,23].

This problem can, however, be approximated by data-driven machine learning algorithms. Using either B- or A-mode systems, several studies have demonstrated US's ability to separate highly similar finger movements[24], shown its robustness against muscle fatigue[25], and demonstrated its applicability as a prosthetic control interface[26–28]. While both signal modalities have shown success, B-mode systems have limited field-of-view[29], high power and processing requirements, and low wearability potential. On the other hand, A-mode systems trade off resolution but can be implemented in a bracelet-like fashion[30], covering the full arm circumference with high wearability and low power requirements[31,32].

Research on A-mode US-based HMIs is largely limited to constrained settings with primary offline validations[33,34], often with participants' arms strapped to eliminate extraneous movements[35]. Real-time usage has been restricted to simple settings with minimal arm movements[18,36], low DoF settings[26], or relied on additional sensors to limit predictions during challenging scenarios[28]. Works so far have also

required participants to produce substantial training data prior to the start of each session. A practical implementation should allow participants to freely move their arms and torso and only require a single initial training session, without recalibration after every donning and doffing event. Ideally, new participants would simply be able to wear the system and start using it immediately, without any prior training.

The main challenge in reaching this realistic implementation is model overfitting. Being an information-rich signal with high spatial and temporal resolution, small changes in sensor or arm position cause large shifts in the data distribution and drops in prediction accuracy[37]. Akhlaghi et al., for example, showed an 10% decrease in accuracy (for 4 gestures) in cross-validations across elbow and shoulder positions, while arm rotations caused such significant muscular changes that control was only possible when training was performed in each position[38]. Research has been carried out to improve performance across donning and doffing events using general models[29,39], methods to reposition sensors accurately[40,41], perform unsupervised model adaptation[42,43], or fine-tune models with labeled data[44]. Yue et al. and Vostrikov et al. were the first to test transfer-learning and fine-tuning, showing that models can also be adapted to new participants by using subsets of data from the new participants, thus restoring performance while reducing computational costs[43,45].

Here we aim to tackling these issues more broadly, building towards a proof-of-concept generic user-independent US-based interface for VR control. We explore models robust to arm movement, large sensor translations and rotations, donning and doffing events across days, and demonstrate strong cross-participant generalization by allowing new users to quickly wear the system and use it with no model recalibration. Our proof-of-concept shows a realistic VR application where participants are able to move their arms freely and interact with virtual objects just like they would in real scenarios. Supplementary Movie 1 briefly summarizes the setup and showcases its control performance. Specifically, we:

1. Develop an easy-to-wear, dry-contact silicone-based A-mode bracelet and used it to collect a large dataset of ultrasound signals paired with joint kinematics during realistic, unconstrained arm movements across four wrist and hand DoFs.
2. Investigate realistic domain shift scenarios and proposed a model training strategy alongside data augmentation pipeline to improve prediction robustness to intra-participant data shifts.
3. Leverage multi-participant training and a referencing method to enable generalization to unseen participants without model fine-tuning.
4. Demonstrate the proposed interface as a proof-of-concept VR controller and proved correlation between its predictions and those of a comparable vision model.

## Results
### Dry, distributed transducer bracelet
We designed a modular A-mode US bracelet made of a series of 3D-printed sensor holders linked by three elastic strings (Fig. 1a). This design allowed high adaptability and 360-degree coverage of the arm in participants with varying arm circumferences (21.8–31.6 cm). Each holder supports a pair of piezoelectric transducers forming two parallel sets of 16 single-element transducers each aligned along the circumference of the forearm. Sensors were placed from the top and held securely in place by a hexagonal protrusion and a plastic cover.

We used a silicone elastomer as the coupling layer between the single-element transducers and the human skin. By pouring, degassing and curing a thin layer directly on top of the sensors, we obtained a coupling comparable to the one provided by traditional US gel (Fig. 1b). However, unlike water-based solutions, silicone is resistant to drying out and can be in contact with the skin for extended periods of time. Previously developed ultrasound gel-coupled bracelets[36]

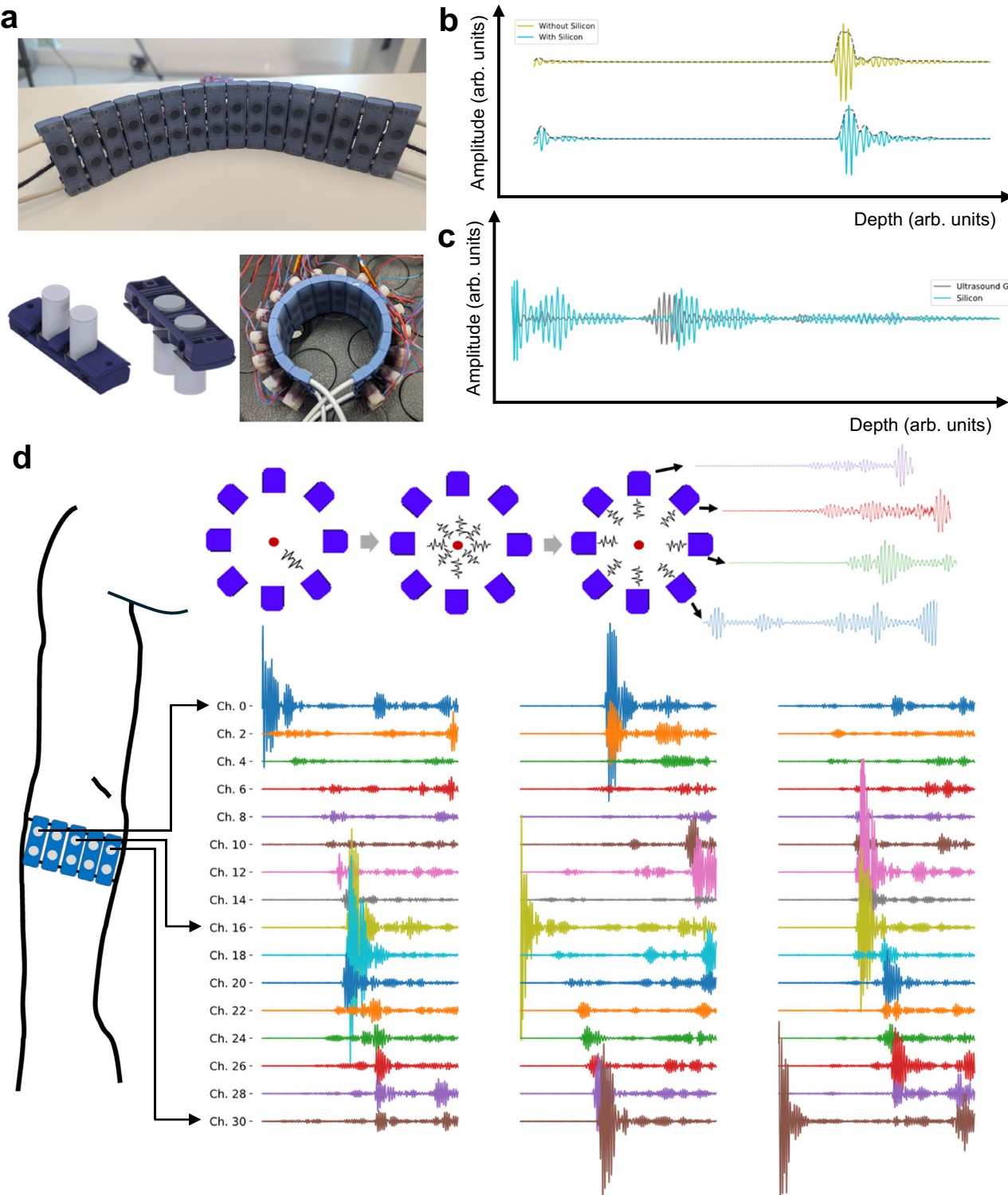

**Fig. 1 | A-mode 32 channel dry-coupling ultrasound bracelet. a** Single-element transducer holder design and assemble in wearable bracelet format. **b** A-mode ultrasound echoes, with and without silicone, in a water bath with a large plastic reflector. With silicone the echo is longer with a small artifact at the end due to internal reflections inside the silicone, but the amplitude is comparable in both cases. **c** A-mode ultrasound echos on a participant forearm aiming at the ulna with and without silicone. Both were recorded at approximately the same position and angle. Similarly to the water bath, the A-mode ultrasound echo with silicone shows

a longer reflection from the bone with a small artifact at the end but only a small difference in amplitude. **d** Illustration of the recording scheme. All transducers transmit one at a time, while every other transducer receives. This is performed for every transducer sequentially, generating a single frame of US scanning. Echoes received by each transducer are therefore dependent on which one is transmitting. In this example raw data from 16 transducers (one row) are seen when, from left to right, transducers 0, 16, and 30 are transmitting.

(including gel application, signal checking, and residue cleaning after use) took more than 5 minutes to put on and take off, whereas the current silicone-coupled bracelet takes less than 1 minute. Thanks to the bottom casing surface, which was specifically designed to reduce peeling and improve silicone adherence, a single application has lasted for multiple months (Supplementary Discussion 1, Fig. S2B).

## Intra-participant generalization

Ten participants ($n = 10$, 3 female and 7 male, $26.8 \pm 2.5yr$) participated in two data collection sessions each generating a dataset of approximately 144 minutes (approximately 24 hours, or 1.1 million samples, in total) of paired US and hand kinematics. Three wrist DoFs (flexion-extension, radial-ulnar deviation, pronation-supination) and one hand DoF (opening-closing of all fingers) were tracked. To understand how different domain shift situations influenced data distribution, recordings were collected in diverse situations. We based recordings on domain shift scenarios likely to occur during real-world usage, including shifts in: (1) upper arm and body position; (2) bracelet rotation; (3) bracelet position; (4) recording session (Fig. 2a).

The dataset was used to develop ML methods robust to these distribution shifts. Fig. 3A shows the performance, expressed as the coefficient of determination $R^2$, of three models (Supplementary Discussion 7, Figs. S8 and S12–S16). The baseline model was previously proposed for small datasets and constrained movements[36]. This model performs poorly when the data is diverse, with a median $R^2$ below 0.5, for all tested generalization groups. Only the pronation-supination DoF performed well in the simpler SameSet and Functional groups with median $R^2$ values of 0.76 and $0.68R^2$, respectively.

The second model leverages the computer vision-based convolutional pipeline proposed in this work (see Methods section for details) that generates A-mode images from multiple echo lines. Similarly to how B-mode images are generated, we recorded multiple views of each transmission event (Fig. 1d). In the bracelet, however, transducers are distributed around the forearm circumference and not side-by-side in an array. Therefore, due to the lack of an absolute and constant spatial relationship between transducers, during a transmission event, the echoes received by the non-active transducers have a variable spatial relationship with each other. Nonetheless, the multiple views of each reflection still encode spatial information. By generating an "image" (Fig. 2c) from each, we impart to subsequent processing steps the intrinsic geometric relationships between the echo lines captured by each transducer. This method performs noticeably better than the baseline, reaching pronation-supination and opening-closing $R^2$ values above 0.9 in the SameSet and Functional groups. Improvements on the generalization groups Position, Rotation and Session were also statistically significant (Supplementary Discussion 3), but still corresponded to poor performance. A similar image formation approach can be performed from standard A-mode single-receive lines. Our results, however, show significantly lower performance when compared to our proposed multi-receive method (Fig. 3B).

Lastly, the third model of Fig. 3A, uses the convolutive model proposed but also implements data augmentation strategies designed specifically for A-mode generated images. Our A-mode images are unlike traditional images and therefore, appropriate domain shifts could not be achieved with standard data augmentation policies. We proposed five policies (Fig. 2d) for data augmentation that act directly on the raw A-mode lines and therefore generate realistic domain shift scenarios. By applying these transformations during training, $R^2$ values for the Position, Rotation and Session groups statistically improved (Supplementary Discussion 3). For hand opening-closing and pronation-supination, for example, $R^2$ performance of the Rotation and Session groups was above 0.9 and comparable to those in the simple SameSet group, confirming the model capacity to achieve similar performance in both seen and unseen bracelet positions.

While generalization across days (Session group) is typically one of the biggest challenges for other HMis approaches[46], our US-based model still maintains high performance across days—likely due to the stability of US signals. The US-based approach appears to have worse performance under shifts in both bracelet position and rotation. The proposed data augmentation methods were capable of simulating realistic rotations (Supplementary Discussion 6, Fig. S7) and heavily improved generalization in that scenario (Supplementary Discussion 8, Fig. S9). Position shifts, however, were harder to generalize, with noticeable lower performance when extrapolating positions beyond its training boundaries (Fig. 3C).

## Cross-participant generalization

With the proposed models validated in single participant scenarios, we then explored cross-session and cross-participant generalization by means of multi-participant models. Both problems are known to be extremely challenging in the brain interfacing and HMI[12] communities. To evaluate US's potential as an user generic interface, we trained models with an increasing number of participants and tested them on both held-out sessions (cross-session, for participants included in the training) and entirely unseen participants (cross-participant). Figure 4a, b show the median $R^2$ performance values of these models for 1 to 9 participants (Supplementary Figs. S4–5 and S17–S18).

Results showed no significant change in cross-session performance with increasing number of training participants (Supplementary Discussion 4), consistent with trends observed in sEMG[12]. For all participants, the most performing model included his US data in the model. For cross-participant generalization however, we found a statistically significant increase in performance, from median $R^2$ of $0.22R^2$ with 1 participant to $0.52R^2$ with 9 participants (Fig. 4a). When focusing at the two best-performing DoFs (Fig. 4b), the results for 9 participants are particularly promising with a small $R^2$ difference of 0.11 between cross-session and cross-participant evaluations, suggesting very strong generalization across participants.

We achieve this strong cross-participant generalization by proposing a referencing method that leverages neutral poses. US can uniquely provide a meaningful morphology reference stage—the sEMG reference stage, for example, is simply the absence of signal and therefore not meaningful. To leverage this, during training, instead of being provided only with the target position, the model received both a reference sample, with the arm in a neutral reference position, and the target position sample (Fig. 4e, see Methods section for details). This can, for a minimal cost, provide a true reference for the model when dealing with never seen participants. Results (Fig. 4c, d) showed that providing the model with a neutral reference significantly increases cross-sectional performance both for models trained with single and multiple participants. Figure 4e, f includes two examples of cross-participant evaluations to showcase the impact on predicted angles. For the cross-participant evaluation the performance gain by providing a reference was comparable to including eight new participants to the training data.

## Validation as a virtual reality controller

Offline results, while useful for comparing models, are poor predictors of usability in realistic scenarios[47]. Therefore, to evaluate the usability of the US interface, the model developed was implemented and tested in real time on a realistic use case. Months after the initial sessions, 16 participants (eight that joined the first sessions, and eight new participants) were invited to wear a virtual reality headset and use a combined optical and US interface to control a virtual twin hand, adapted from the DARPA Haptix project[48], to interact with a virtual environment (Fig. 5a). No data collection was performed during this session (zero training) and the model was used without fine-tuning. Participants completed three repetitions of five tasks in a complex environment with simulated contact physics (see Methods section for details).

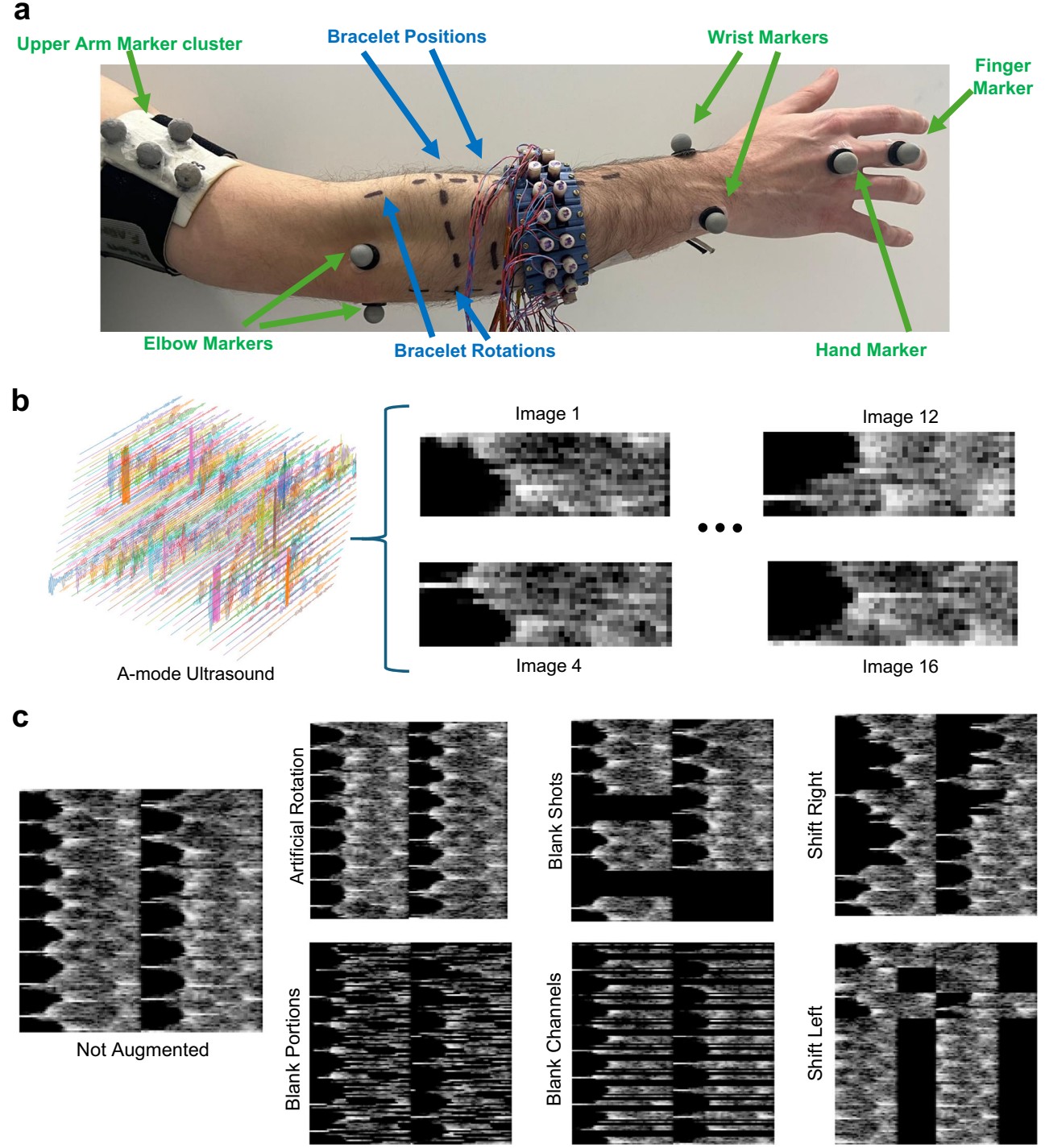

**Fig. 2 | Data collection, image formation and augmentation. a** Ten passive reflective markers were positioned on the right arm of each participant. Four markers belonged to a upper arm cluster, two markers at the elbow on the styloid process of ulna and styloid process of radius, two markers at the wrist on the medial epicondyle and lateral epicondyle, and two at the hand on the head of the metacarpal bone of the middle finger and head of proximal phalanx of the middle finger. Data was collected with the bracelet at three distinct positions and 4 different rotations. **b** Raw A-mode multi-receive data from each circle of 16 transducers was collected and processed into a collection of 16 A-mode images per frame. **c** Examples of a non-augmented frame and the same frame augmented with the 6 proposed methods. From top to bottom and left to right, artificial clockwise rotation of channels, zeroing of random continuous amounts of A-mode data, zeroing of transmissions, zeroing of channel recordings, lateral right shift of shots, lateral shift to the left of shots.

For example, users were presented with a virtual mug and ball, asked to reach for the ball, grab it, carry it, and drop it inside the mug. Mimicking realist scenarios, strict end stages were not enforced and participants were allowed some freedom in how they approached tasks. Supplementary Movie 1 illustrates some exemplary trials.

To support evaluation of overall system performance against comparable alternatives, the participants' movements were also captured by a compact commercial vision-based motion capture system (Leap Motion). Based on the landmarks predicted by the optical system the equivalent wrist and hand DoFs were calculated

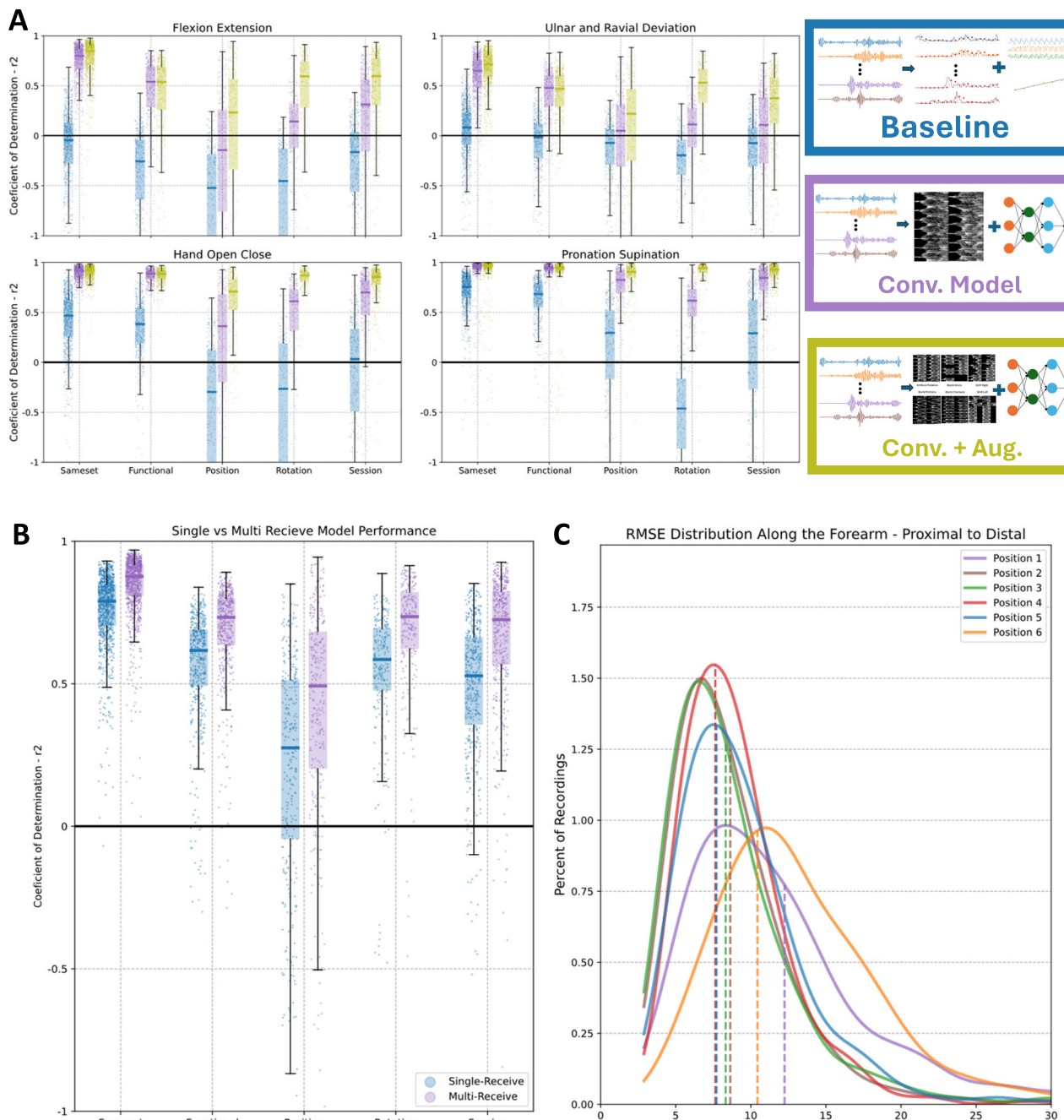

**Fig. 3 | Intra-participant models results. A** Box and whiskers plot with scatter points of $R^2$ between prediction and ground truth for the baseline model (Baseline, blue), proposed convolutional model (Conv. Model, purple), and convolutional model with data augmentation (Conv. + Aug., yellow) separated by degree of freedom. Single participant models did not use the neutral referencing method. Results are for training/testing splits for each data shift group evaluated, including data from the same sets (SameSet), from different upper arm positions (Functional), armband positions (Position), armband rotations (Rotation), and sessions (Session). Scatter values bellow −1 were not included for visualization purposes.

The convolution model drawing is merely illustrative and was made with[81]. **B** Box and whiskers plot with scatter points of $R^2$ results between two models. First, in blue, uses a standard A-mode single-receive strategy while the second, in purple, uses the proposed multi-receive method for image formation. Scatter values below −1 were not included for visualization purposes. **C** Kernel density estimations of the *RMSE* results for training/testing split of different positions with the model trained on all positions and tested on a single held out position. Doted line marks the median of each distribution. In order, position 1 is the most proximal (closer to elbow) and 6 is the most distal (closer to hand).

and compared against the US system predictions. Unlike our offline results, this was not a true performance evaluation—as the Leap Motion is known to have limited tracking accuracy[49,50]—but a comparison between two similarly portable systems. Supplementary Movie 1 also includes examples of vision and US predicted trajectories while Fig. 5b quantifies the agreement between both systems via the coefficient of correlation. Results showed high correlation

between the predictions for the pronation-supination and hand opening-closing DoFs, a medium (and highly variable) correlation for the flexion extension DoF, and low correlation for the radial and ulnar deviation. Lastly, even in a limited workspace with no objects, the Leap Motion failed to localize the hand and provide predictions for 1.5 ± 2.3% of the total frames, with up to 12% missing predictions in a single recording.

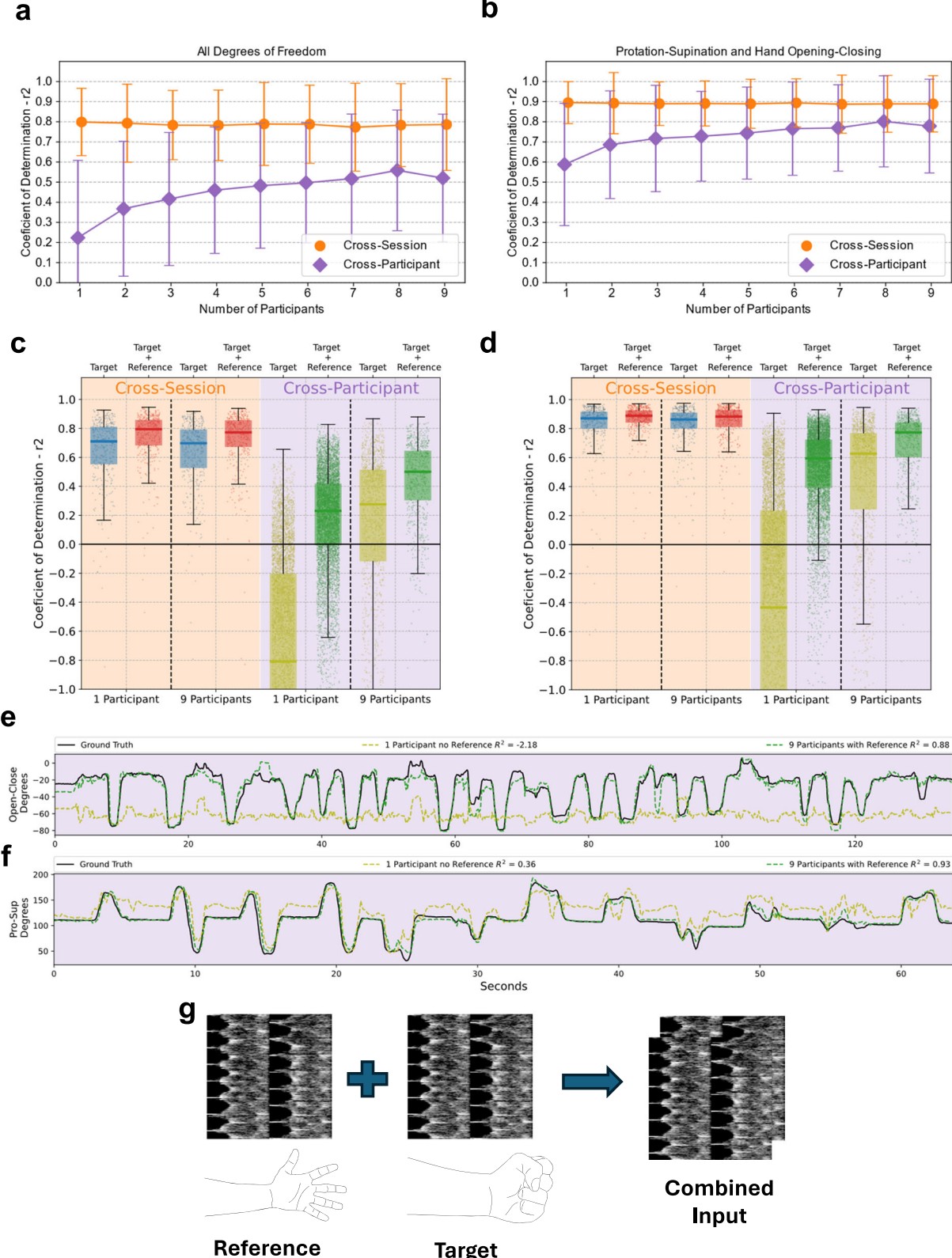

## Discussion

Robustly tracking upper limb kinematics is instrumental for natural engagement with XR experiences. To date, no technological solution combines wearability and robustness in diverse environments. We proposed a dry US-based A-mode armband that can be easily donned and doffed. This design allowed us to collect a diverse dataset of over 24 hours of labeled US data. Leveraging this dataset and modern supervised deep learning methods, we investigated intra-session, cross-session, and cross-participant dataset shifts, while proposing methods capable of robust generalization in each situation. With the final models, we implemented a virtual environment and demonstrated robust control and high usability by naive participants.

Multiple A-mode bracelets have been proposed[30,39,51–53], but all have used standard US gel as the coupling medium. Nonetheless, the

**Fig. 4 | Multi-Participant Models and Cross-Participant Generalization.**
**a**, **b** Categorical line plots of multi-participant, from 1 to 9 participants, models $R^2$ performance. Each point represents the median of the $R^2$ distribution of each recording tested for the four DoFs (**a**) and only the pronation-supination and hand opening-closing DoFs (**b**). Error bars represent the distribution standard deviation. Includes results for cross-session (circle, orange) and cross-participant (pentagram, purple) evaluations. **c**, **d** Box and whisker plot with scatter points of $R^2$ results for each recording. Scatter points represent the median $R^2$ across the four DoFs for each recording tested for the four DoFs (**c**) and only the pronation-supination and hand opening-closing DoFs (**d**). Results are separated between cross-session and cross-participant validations with pairs of models trained without and with (Target + Referencing) and without (Target) the referencing strategy. Pairs are shown for models trained with only one participant and with 9 participants. The y axis was cropped at −1 for visualization purposes. **e**, **f** Examples of cross-participant validation recordings with predictions from the models with 1 participant and no reference and from 9 participants with reference. A hand Open-Close DoF wrist recording of participant five (**e**) and a Pronation-Supination DoF functional recording of participant seven (**f**). **g** The proposed model combines a reference neutral position frame (hand relaxed with the ulnar styloid pointing downwards) with the target position frame. Both are combined in a two channel image used as input to the convolutional model.

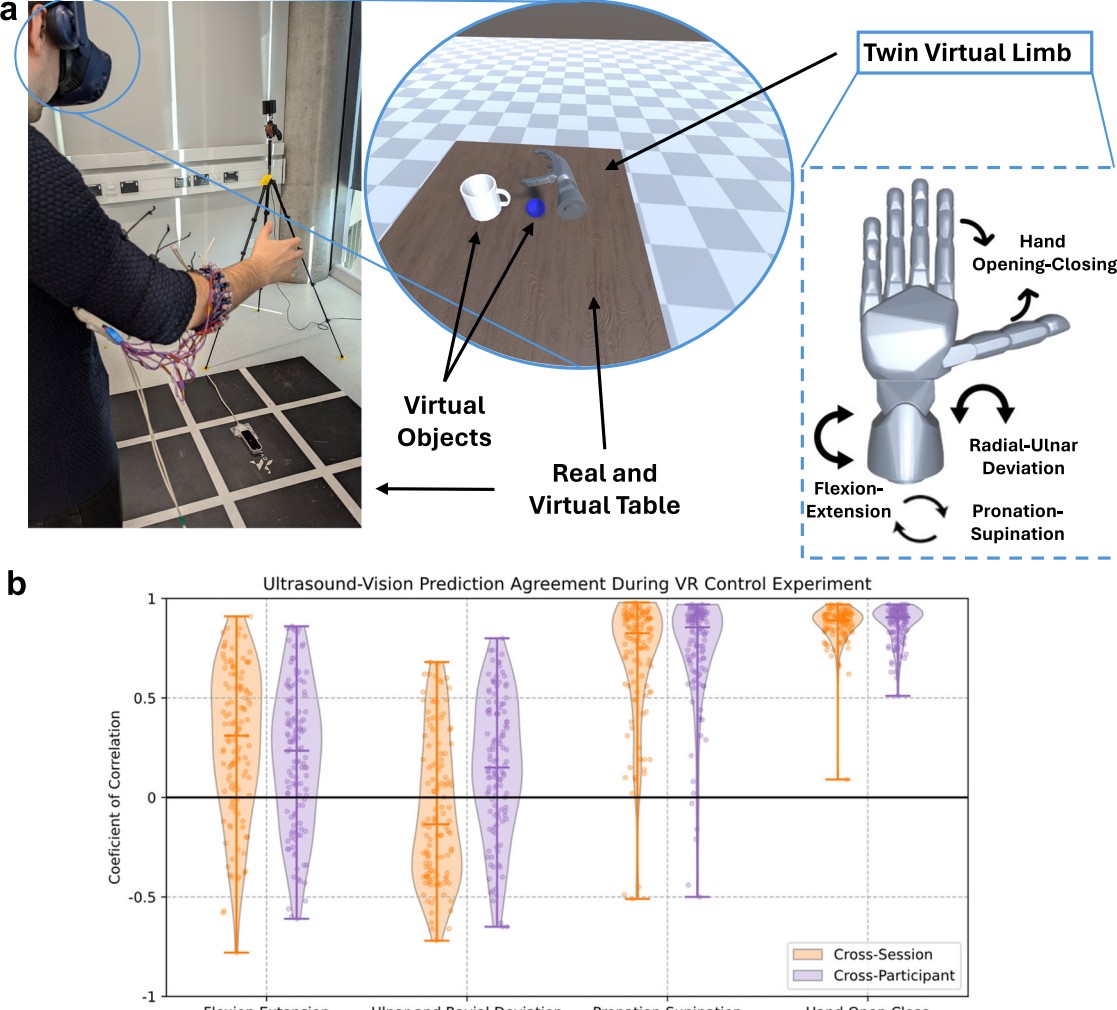

**Fig. 5 | Virtual reality control demonstration. a** Participants were presented with a realistic virtual environment with simulated contact physics. Using the US-based controller and optical system they were asked to interact with virtual objects and perform five different interaction tasks. The US armband controller allowed smooth continuous simultaneous control the wrist flexion-extension, radial-ulnar deviation and pronation-supination as well as the hand opening-closing. **b** Violin and scatter plot of the agreement between joint predictions, for all participants, from the US system and the optical system. Results are expressed as the Person correlation coefficient between predictions during each of the recordings. Results are separated between participants that participated in the data collection sessions (Cross-Session, orange) and unseen participant with no fine-tuning (Cross-Participant, purple).

unsuitability of US gel as a viable coupling method for real-world usage has been discussed in the literature[29] and alternatives have been proposed. For example, some wearable US patches have leveraged encapsulated hydrogels as more stable interfaces[54]. Hydrogels maintain ideal coupling capabilities while providing longer-term usage (up to multiple days) but are fragile and brittle, and may also cause skin irritation[55]. Other patches have used silicone elastomers for transducer encapsulation and silicone adhesives for skin interfacing[56,57]. While silicone interfaces suffer significant impedance mismatch with tissue, they are inert, biocompatible, and resistant. Inspired by these works, we have proposed a similar methodology for coupling A-mode transducers for HMI applications. Due to the requirement of multiple cycles of donning and doffing, unlike patches that are normally single use, we proposed to use a softer silicone interface, with very low shore hardness, that can provide adequate encapsulation and skin interfacing. We have used a single silicone encapsulation for upwards of 4 months with little visible damage to the silicone coupling surfaces (Supplementary Fig. S2) and no apparent change in signal quality.

Large, diverse, and high-quality datasets of labeled data are one of the key factor in unlocking more generalizable models. Compared to other US-based HMI works, the dataset collected is substantially larger (~30x[52], 20x[53], 10x[36], and 3x[39]), and notably diverse, uniquely incorporating multiple arm positions, bracelet positions, rotations, and sessions. Against sEMG datasets, our dataset is medium-sized when compared to academic works (~4x[58], 1.6x[59], and 0.3x[60]), but relatively small when compared to recent industry efforts (~0.03x[2], and 0.0015x[12]). Given our dataset size, the demonstration of robust generalization and resulting consistent control was surprisingly strong and points towards promising potential for US-based interfaces.

Compared to other US literature, our offline results demonstrated that the proposed models were capable of fitting individual participant data. Our results for the SameSet group are comparable to Guo et al.[21] ($0.9R^2$) for wrist flexion-extension and Yang et al.[52] ($0.95R^2$), as well as the average 7° RMSE results for three DoFs from Spacone et al.[39], and 5.3° for the flexion-extension DoF from Iravantchi et al.[61].

Our results for the other cross-validation groups have less clear comparisons in the literature. Changes in sensor position have been reported by Kamatham et al.[62] with large drops from $R^2$'s above 0.95 to below 0.2 for small 2.4 mm scanline shifts in some cases. On the other hand, Vostrikov et al.[43] have shown robust performance of 96% in 4-class classification accuracy when exploring small bracelet rotations (≈30°) and translations (≈3 cm). Our tests expanded on these by showing models robust to complete full rotations and position shifts of more than 5 cm.

Performance across sessions has been shown to also be a challenge, with Khan et al.[63] and McIntosh et al.[29] building models somewhat robust to it, but showing notable performance degradation when slight probe shifts were present. Yang et al.[64] and Shangguan et al.[40] have separately proposed methods to help users reposition sensors to the same locations, improving classification accuracy by 23% over six classes with B-mode US and by 30% over ten classes for A-mode, respectively. Lykourinas et al.[42] also showed the impact of intra-session evaluation, with models losing over 50% of their classification accuracy, but proposing that unsupervised adaptation using data from the new sessions can restore accuracy by 25%. Lastly, the results from Spacone et al.[39] are the most comparable to ours in the context cross-session validations, with RMSE results slightly worse when compared to ours (10.4° vs 10.1° on the flexion-extension DoF, 9.4° vs 5.8° for radial-ulnar deviation, and 13.5° vs 8.5° for hand open-close).This highlights the advantages of our approach, as our recording setting included more challenging free arm movement and a fourth DoF (pronation-supination).

As far as we are aware, no other work has developed a US-based model capable of cross-participant generalization without model recalibration. Two works have, however, explored cross-participant prediction by retraining models. Lian et al.[45] employed weight fine-tuning with an unspecified amount of data from new participants, and showed that performance could be fully recovered in most cases, while computational time can be reduced. Similarly, Vostrikov et al.[43] used transfer learning, showing that while the classification head of a model needs to be trained for each participant, the feature extraction portion can be borrowed from other participants while maintaining performance. Our results point to a more generalized direction showing that cross-participant evaluations can be performed with US data—without any model adaptation. By combining models trained over multiple participants with the natural ability of US to provide a neutral reference, we show that models are capable of robust performance in the previously unseen arm morphologies of new participants. Surprisingly, we show that performance for the pronation-supination and hand open-close DoF is similar both for cross-session and cross-participant evaluations, hinting that much of what the models have learned for these DoFs is independent of individual participant anatomies, but is likely tied to generalizable patterns in muscle activation. It has recently been shown that the sEMG HMI problem mirrors other machine learning problems with an inverse power law performance scaling with model and dataset size[12]. Our results include a maximum of 10 participants and therefore cannot demonstrate the same applies for US interfaces; however, the high performance increases in the narrow 1 to 10 participant range point towards some continued scaling with larger datasets.

With sEMG, robust finger joint angle prediction, under constrained conditions, has been explored by Liu et al. with a mean error of less than 10° but large variance[59]. Simpetru et al., on the other hand, had high mean $R^2$ values of approximately 0.84 for simple movements but significantly lower values (≈0.14) for random movements[58]. Recent industry-led efforts into large dataset acquisitions have expanded this to more general scenarios, achieving finger joint angle mean errors of around 15°[2]. However, they also argue that pose regression (predicting joint angles from sEMG alone) is only a partially observable task and therefore particularly challenging with sEMG being more appropriate for pose tracking (predicting joint angles, for a short time, from sEMG signals, assuming a known initial position). It is unlikely that US can be characterized in the same way as its information is directly related to kinematics. Using these nomenclatures, our proposed referencing method could be seen in parallel to an sEMG pose tracking task. But instead of relying on an external system to constantly provide a reference state, our US approach uses a internal reference that only needs to be updated once per session.

Our real-time VR validation is also distinct within the US literature. The most complex online evaluations of US-based systems involved multi-DoF target achievement control (TAC) tasks[36,41,52] and functional prosthesis control[26–28]. For sEMG, a large corpus of TAC studies exists[65,66], with some works including real-time validation in simple VR environments that do not include contact physics[67,68]. The works by Kumar et al.[48] and Chappell et al.[69] have served as the key inspirations and references for our VR experiment implementation. Kumar et al. integrated a CyberGlove with MuJoCo contact physics, while Chappell et al. integrated sEMG signals, both enabling realistic and dexterous interactions in VR.

Unlike our offline evaluations, our VR validation did not strictly measure control performance. Any performance metric extracted from these sessions would inevitably be dominated by challenges associated with the VR environment as none of our participants had prior experience with environments featuring realistic physics, and over 60% had little to no VR experience. We therefore opted to compare the US system predictions with the predictions from a comparable vision-based system. The Leap Motion was chosen both as a robust system for wrist 3D position tracking and as a comparable interface. The Leap Motion has been validated against gold-standard marker-based optical tracking systems with varied but mostly acceptable results both for the wrist[50] and the fingers[49]. We also highlight that perfect prediction accuracy is not necessary for effective real-time usage of HMIs as robust, but inaccurate predictions, can still enable user adaptation and result in good performance[47].

Overall, the US and the vision system showed high correlation for the pronation-supination and hand opening-closing DoFs. This aligns with our observations, as all participants demonstrated robust control over these two DoFs. In most cases of low correlation between vision and US predictions for pronation-supination, it was clear from the videos that the vision system was wrong, consistent with prior studies reporting poor pronation-supination tracking by the Leap Motion[50]. Flexion-extension showed mixed results, with high variability in correlation between predictions. Videos showed cases where either one or both systems produced visibly incorrect predictions. Radial-ulnar deviation generally showed very low correlation between predictions, due to low usage of the DoF during the tasks and unsatisfactory tracking by the US system. It was observed that while tracking was stable, most participants demonstrated limited control over this DoF.

Lastly, we highlight some limitations of the work developed and paths for potential further research. First, a clear limitation for the work involves the performance of the radial-ulnar deviation DoF (Supplementary Discussion 9, Fig. S10). Participants in the VR evaluation experiment had limited control over the radial-ulnar deviation movement and range of motion was normally limited to less than 50%. Its unclear if this low performance is a limitation of the US modality or due to lack of a sufficient dataset or limitations of our setup. Exploration of this with different bracelet positions (more distally along the forearm, closer to the wrist), higher transducer count or higher transducer center frequency could prove to lessen this issues. While the dataset collected is large compared to previous US works, it is still small compared to what other areas used to achieve very robust control, and the tested participant distribution was narrow. Following recently developed flexible transducers, new methods for beamforming data when transducers deform have been proposed[70]. By leveraging these approaches, echoes from our bracelet could possibly generate a single cross-sectional image of the forearm, likely improving the robustness of the control methods. Another approach would expand on the concepts of acoustic interferometry proposed by Iravantchi et al. that leverage multi-transducer transmissions to extract more fine grained information from users' arms[61].

The current proof-of-concept predicts four upper limb DoFs, and achieves complete coverage of the workspace by tying predictions to a exocentric reference based on a optical MOCAP system. While a standalone upper-limb US interface could be implemented by fixing joint predictions to a egocentric reference, based on the user's field-of-view, natural usage would require prediction of elbow and possibly shoulder joints. A similar approach has been recently demonstrated[71] and could be implemented in our setup by splitting the bracelet into two, one for the forearm and one for the upper arm. While its likely that elbow joint flexion-extension angle can be robustly predicted by our methods, whether robust predictions of shoulder joint angles are possible from upper arm morphological deformations is an open question that should be explored by future works. Lastly, the current work used a portable, but not wearable system to drive and record the US signals[72]. While most VR systems still require a dedicated desktop, future work should expand the setup to fully wearable electronics[31,32].

In conclusion, we developed a US-based proof-of-concept HMI capable of robust cross-participant predictions. Leveraging a dry-coupling and easy-to-wear armband we recorded a large dataset of paired US and kinematics and used it to explore deep ML methods robust to intra-session, cross-session and cross-participant shifts. We also implemented the developed models in a real time VR control environment and demonstrated that users were capable of robust and accurate control. Our proof-of-concept builds a solid foundation for realistic usage of US-based control by control. The robust offline prediction results and the reliable real-time control trials give solid indication of US's potential as an HMI signal, strongly supporting its use in the next generation of HMIs.

## Methods
### Participants
Ten healthy participants (*n* = 10, 3 female and 7 male, 26.8 ± 2.5*yr*, 27.0 ± 2.7*cm* forearm circumference) were recruited for the first two sessions of data collection. Eight of the initial ten participants (*n* = 8, 2 female and 6 male, 26.6 ± 2.8, 26.6 ± 2.7*cm* forearm circumference) participated in the third session of VR control. Eight new healthy participants (*n* = 8, 1 female and 7 male, 30.3 ± 5.7, 26.9 ± 2.7*cm* forearm circumference) were also recruited for the third VR control session, for a total of 16 participants joining the third session. Details are enclosed in Table 1. All procedures were approved in accordance with the Declaration of Helsinki by the Imperial College Research Ethics Committee (ref: 22IC7602). Before data collection, participants were

**Table 1 | Participants information and Session Separation**

| ID | Session 1 & 2 | Separation Session 1 & 2 | Session 3 | Separation Session 2 & 3 |
|---|---|---|---|---|
| P01 | Y | 1 day | Y | 233 days |
| P02 | Y | 1 day | Y | 224 days |
| P03 | Y | 5 days | Y | 220 days |
| P04 | Y | 5 days | N | - |
| P05 | Y | 9 days | Y | 182 days |
| P06 | Y | 3 days | N | - |
| P07 | Y | 1 day | Y | 177 days |
| P08 | Y | 5 days | Y | 114 days |
| P09 | Y | 2 days | Y | 107 days |
| P10 | Y | 5 days | Y | 100 days |
| P11 | N | - | Y | - |
| P12 | N | - | Y | - |
| P13 | N | - | Y | - |
| P14 | N | - | Y | - |
| P15 | N | - | Y | - |
| P16 | N | - | Y | - |
| P17 | N | - | Y | - |
| P18 | N | - | Y | - |

briefed on the study, presented with a participant information form, and asked to sign a consent form.

### Silicone coupling layer
Silicone elastomer was cured directly on each holder supporting a pair of piezoelectric transducers. Sensors were placed from the top and held securely in place by a hexagonal protrusion and a plastic cover with two screws. With both transducers in place and the cover tightly screwed into each holder, the band was flipped upside down and taped securely to a table so silicone elastomer (shore hardness 000-34, EchoFlex Gel 2, SmoothOn) could be poured over each and left to cure overnight. Before pouring, parts A and B of the silicone were thoroughly mixed and left to degas at 0.8*bar* for approximately 2 minutes. This allowed for most of the air to escape from the mixture but not for enough time to pass for it to start thickening, allowing a clean and even pour over the holders. Other elastomers with higher shore hardness were tested but were found to only provide appropriate coupling with the skin by either applying adhesive or excessive pressure. For our soft silicone coupling method we empirically found that a thickness of ~1.5 mm worked best (Supplementary Discussion 1). During the course of experiments, we identified that even with the very soft silicone some participants required slight adjustments of the bracelet until a good coupling was reached. We believe that this could be related to skin dryness, and found that by using a small amount of common skin moisturizer prior to positioning the bracelet excellent coupling could always be achieved.

### Transducers and acquisition hardware
For our experiments, the armband was composed of a series of 16 holders (with 2 transducers each), totaling 32 single element transducers (Fig. 1a). We used 1-MHz piezoelectric transducers (circular, 2.5 mm radius, 15° opening angle) (Supplementary Fig. S1) connected via twisted pair cables to a portable acquisition system (MoUSE, Fraunhofer IBMT, Sulzbach, DE)[72,73] controlled by a custom-made graphical user interface. Transducers were driven by a 40*V* square biphase three-cycle burst pulse. The MoUSE system both drives and records US acquisitions in arbitrary transmission sequences at 12-bit resolution and 50 MHz sampling rate. To record US data, we employed

a single transmit multiple receive strategy[36]. Each frame of US comprises 32 sonification events (one for each transducer, in order) acquired one after the other. For each sonification event, one transducer acts as a transmitter and the same transducer, as well as the rest of the transducers, acts as a receiver with echoes being recorded for $110\mu s$. Therefore, each frame consisted of $32 \times 32 \times 5400$ samples. The MoUSE system was connected to a computer via USB 3.0 and was programmed to wait until the current frame was completely transmitted to the host computer before acquiring a subsequent one. On our setup, frames were acquired at approximately $11.8 \pm 0.5 Hz$ (Supplementary Discussion 2, Fig. S3).

To record wrist and hand kinematics, a marker-based optical motion capture setup was used (Vicon Nexus, 28 cameras in total with 16 Vantage 8 and 12 Vero v2.2 cameras, Oxford Metrics Ltd, Oxford, UK). 3D marker $x$, $y$, $z$ coordinates of $14mm$ passive reflective markers were collected at $200 Hz$. Six markers were positioned at specific forearm and hand landmarks and four were attached to a band strapped around the upper arm. Marker placement followed anatomical landmarks per palpation guidelines[74] using a double-sided hypoallergenic tape. Marker placement is described and illustrated in Fig. 2a.

US and motion capture recordings were synchronized an Arduino. At the start of US recording, the Arduino generated a synchronization trigger that started/stopped the motion capture system (Vicon Lock Lab, Oxford Metrics Ltd, Oxford, UK). Each recording was then manually checked and corrected when needed to ensure the start of the first movement matched in both the US data and motion capture data. Frames from the US and motion capture system were them matched, during model training, based on the time from start of the recording. The variable US timestamps sent by the MoUSE were matched to the stable timestamps from the motion capture samples acquired by the Vicon system.

## Ultrasound preprocessing and image formation

The collected US data was preprocessed in order to generate US images from the A-mode lines. Before image formation, each $32 \times 32$ channel frame was broken into two separate $16 \times 16$ frames (the two off-circle $16 \times 16$ frames were discarded). This was done because the bracelet was designed with 2 identical circles of 16 transducers, so for processing we considered each to be distinct. In practice, this allowed the collection of two unique samples for each frame, effectively doubling the amount of data collected. For each frame, one image was generated for each transmission and its respective 16 receiving events, for a total of 16 images per frame (Fig. 2b). Images were generated following a preprocessing pipeline based on standard B-mode image formation. Initially, A-mode lines were hardware time gain compensated via an 8-stage amplifier manually tuned to compensate for silicone and soft tissue attenuation. For each A-mode line of 5500 samples, the first 800 were removed to exclude static noise artifacts and the signal from reflections inside the silicone coupling layer. After digitalization, data was band-pass filtered between 0.4 and 1.6 MHz, Hilbert enveloped (with reflective padding at the edges), log-compressed into dB, and axially downsampled by 100. For the offline experiment, data was saved in the raw frequency format and processing was applied after the fact. For the online virtual reality experiment processing was applied on the fly (Supplementary Discussion 2, Fig. S3).

## Data collection

Participants were invited for two identical data collection sessions. The two sessions were completed on different days, with at least 24 hours between each other (Table 1). On each, participants were asked to stand inside the motion capture volume in front of a large monitor. The flexible armband was positioned on the participant's right forearm by an experimenter and the individual sensor holders were loosely arranged to cover the entire forearm circumference. Two sets of short

videos were shown to the participants on the screen and participants were asked to imitate the motions with their wrist and hand. The order of the short videos on each set was randomized to keep participants engaged, enhance variety in the participants' motion, and avoid memorization. Participants were explicitly told to be relaxed, perform all movements naturally, and not worry about maintaining a consistent position during the experiment. Participants were also told not to worry about making mistakes when following the videos, as the videos were only used to guide participants in exploring varied movement combinations. Labels were only derived from the optical motion capture system.

The first set (labelled Wrist Set) was composed of 26 short, 5 s videos (totaling approximately 2 minutes) showing a hand performing wrist and hand movements. Videos always started with the hand in a neutral position (hand relaxed with the ulnar styloid pointing downwards, upper arm relaxed to the side and elbow flexed around 90 degrees) and involved moving one (or more) DoFs of the wrist or hand to its maximum and then going back to neutral. All movements included only the opening and closing movement of all fingers together and not individual motion of fingers.

The second set (labelled Functional Set) was composed of 24 different 4-second videos, with only half (12) shown in each repetition (totaling approximately 1 minute). These videos showed a hand performing functional movements in an imaginary virtual reality setup. Movements included combinations of wrist, hand, elbow, upper arm, and trunk movement. For example, videos included moving imaginary objects from an elevated position to the floor, slicing imaginary objects, or rotating imaginary valves.

After participants completed both video sets, the experimenter repositioned the bracelet, and participants repeated the videos. The bracelet positioning followed 12 predefined positions (Fig. 2a). Before the first repetition, a line was drawn from the antecubital fossa to the wrist. Based on it, three more lines (parallel to the first one) were drawn representing a 90-, 180-, and 270-degree rotation from the first line. These lines defined four different rotations for the bracelet. Following that, three more lines, perpendicular to the first four lines, were drawn around the forearm with the first ~3 cm distally from the antecubital fossa and the following two at around 1.5 cm distance distally of the previous. These lines defined three positions for the bracelet. For most participants, no positioning marks remained from the first session. As a result, the positional lines had to be redrawn. While the new positions were likely similar to those in the first session, they were not guaranteed to be identical.

In total, each participant had 48 recordings (4 rotations × 3 positions × 2 video sets × 2 sessions), or approximately 72 minutes of US data. At approximately 11 frames per second, this resulted in around 50,000 frames per participant. With our double circle recording approach, we double this to approximately 100,000 samples (or 144 minutes per participant). With 10 participants, we reach approximately 1.1 million unique training examples. In practical terms, our dataset (in number of images) is similar in size to the widely used computer vision ImageNet-1k dataset[75], which contains ~1.5 million images.

To separately explore generalization capabilities over distinct intra-participant and cross-participant situations, we have defined six specific cross-validation groups. Each group aimed at exploring one specific facet of generalization. The first four groups were employed only for the single-participant models, the fifth group was used for both single- and multi-participant models, and the sixth group was used only for multi-participant models.

1. The SameSet group used the first 70% of all recordings as training data and the last 30% as testing data. It aimed at representing the simplest possible situation with all tested positions being present in the training data. We, however, highlight that due to the fact

that each video was only played once per repetition, the testing dataset was still composed of out-of-set movements that were not present in the training set (at least not in the same bracelet position) making even this simple case likely more challenging than cases normally found in the US literature. This was the only group where individual recordings were split between training and testing.

2. The Functional group used all recordings from the Simple Set of videos as training data and all recordings of the Functional Set videos as testing data. It aimed at exploring model generalization at different forearm, elbow, and trunk positions.

3. The Position group used all recordings of two of the bracelet positions (and all rotations) as training data and the other third position as testing data. All three combinations of positions were tested. It aimed at exploring model generalization for previously unseen bracelet positions.

4. The Rotation group used all recordings of three of the bracelet rotations (in all positions) as training data and the other fourth rotation as testing data. All four combinations of rotations were tested. It aimed at exploring model generalization for previously unseen bracelet rotations.

5. The Session group used all recordings from one session as training data and the recordings from the other session as testing data. It aimed at evaluating cross-session generalization by incorporating dataset shifts due to nonstationarities as well as realistic shifts in position and rotation. This cross-validation group was used in both the single- and multi-participant evaluation sections.

6. The Participant group used all recordings from one or more participants as training data and the recordings for other participants as testing data. It aimed at evaluating the most challenging scenario possible, simulating performance on a completely unseen cross-sectional morphology. This cross-validation group was used only on the multi-participant models evaluation section.

## Motion capture preprocessing

The recorded 3D marker coordinates were first manually preprocessed and cleaned by an expert using the Vicon Nexus software. Small gaps in the markers trajectories (less than 20 samples) were automatically interpolated by a spline algorithm, while larger gaps were manually interpolated by an expert using nearby donor marker trajectories. The study intentionally used as few markers as possible to make the cleaning process simple, robust, and not time-consuming. Specifically, the four markers in the upper arm cluster (Fig. 2a) were only used as donor markers to correct gaps on other markers' trajectories. Using the cleaned 3D marker coordinates, angles for the three wrist DoFs and the hand opening and closing DoF were calculated using inverse kinematics models. Wrist flexion-extension and radial-ulnar deviation followed the Vicon Plug-in Gait model[76], with angles calculated between the hand plane (defined by the radial and ulnar styloid and the distal end of the middle finger metacarpal bone) and the non-rotating forearm plane (defined by the radial styloid, the midpoint between the radius-ulnar styloids, and the midpoint between the medial and lateral epicondyle). Wrist pronation-supination was calculated in a similar manner, but used a forearm plane anchored at the elbow (defined by the medial epicondyle, lateral epicondyle, and midpoint between the radial-ulnar styloids)[77]. This definition for wrist pronation-supination avoided the need for markers on the upper arm, simplifying the recording setup and improving tracking robustness. The angle between the plane of the proximal phalange of the middle finger (defined by the proximal and distal ends of the phalange) and the palm of the hand (defined the same as above) was calculated and used as the hand opening and closing angle. Before being used for model training, angles were normalized by the range of motion measured in each recording session. Therefore, the labels used were not raw angle values, but normalized DoF activation values. Our results point

towards the fact that session normalization improves model performance (Supplementary Discussion 10, Fig. S11).

## Data augmentation

Appropriate data augmentation is crucial for proper training of generalizable vision models. Due to the uniqueness of our problem and data recording setup (the bracelet setup, the used multi-receive strategy, and the image formation method), traditional data augmentation policies used in computer vision would not be appropriate in generating suitable domain-shifted examples (Supplementary Discussion 6). In our situation, to properly approximate realistic domain shift changes, the data augmentation policies needed to affect the raw A-mode data before image formation. We therefore developed and proposed the usage of two specific policies (Fig. 2c).

First, we propose the use of an artificial rotation policy where channels are rotated clockwise. For each example, all channels (and corresponding receive events) are rotated clockwise by a factor of $N$ channels, with $N$ being a random number between 1 and 15. This method allows for the realistic simulation of bracelet rotations, without needing to physically rotate it and rerecord the trial (Supplementary Discussion 5, Fig. S6).

Second, we propose the use of specific blanking and shift policies that act on the A-mode channels and not on the images. In our implementation, for each example, one of 5 augmentation strategies (or none) was applied.

1. Blank Shots. The image can have a random number of shots (and their corresponding receive events) between 1 and 8 blanked.

2. Blank Channels. The image can have a random number of channels (in all transmit events) between 1 and 8 blanked.

3. Blank Portions. Randomly distributed portions of data (of varying size) can be blanked from all channels.

4. Shift Left. All channels can be randomly shifted to the left by up to 200 samples.

5. Shift Right. All channels can be randomly shifted to the right by up to 200 samples.

## Models and model training

All models were built on PyTorch and trained on single GPUs on a High-Performance Cluster. Before training, each sample containing 16 $16 \times 470$ images had a sensitivity cut-off of -50 dB applied and was converted to pixel values (from 0 to 255). Following, the 16 images were combined into a single image (two rows of eight images each) of $128 \times 940$ pixels. Lastly, images were resized to $128 \times 128$ pixels, min-maxed between 0 and 1, and z-score normalized using the approximate dataset mean and standard deviation.

In the experiment comparing the single-receive strategy, images were formed just from the 16 A-mode channels, following the same process (but with a -40 dB sensitivity cut-off). The baseline model tests followed exactly the same process as described in ref.[36].

We empirically experimented with a range of modern convolutional network backbones (e.g., ResNet, DenseNet, ViTs, SwinT). Overall results followed similar trends regardless of the backbones with the best-performing architecture being the ConvNeXt[78]. We experimented with the model sizes defined in the original work (Tiny, Small, Base, and Large) and selected the Tiny due to similar performances across sizes and ease of training and testing with the smallest model. It is likely that an even more diverse dataset is needed to leverage larger models appropriately. The standard architecture was modified so that the first layer would expect one-channel gray-level images (or two channels when using our neutral referencing method) and the head was replaced with a fully connected two-layer regression head (input-output: 1024-100 and 100-4).

We also proposed a method to leverage a US reference stage. Besides being provided with the target sample, models were provided with a reference US sample of the participant' arm in a rested position

**Table 2 | Virtual reality tasks description**

| Task Name | Objects | Task Descriptions |
|---|---|---|
| Ball and Mug | Movable Ball and hollow mug | Approach and grasp the ball, move it to the hollow mug and drop it inside |
| Cylinder Relocation | Cylinder and base with two holes and a separation | Approach and grasp the cylinder, move it over the separation and release it on top of the second hole |
| Door Handle | Solid wall attached to a rotating door handle | Approach the handle and grasp it, rotate the handle clockwise halfway, rotate back to starting position and release |
| Liquid Pouring | Hollow cup with 5 small circular beads inside and empty solid hollow cup | Approach and grasp the cup on the right (with the beads inside), lift the cup over the top of the second cup and pour the beads from one cup to another, return to starting position and release the cup |
| Shelf Stocking | Solid horizontal shelf and a movable rectangular plank | Approach and grasp the rectangular plank, adjust the plank position (by rotating and extending the hand), slide it inside the shelf and release |

(hand relaxed with the ulnar styloid pointing downwards). This was performed by simply adapting the first layer to receive a two-channel image with the first channel being the reference neutral sample, and the second the target sample. During training this was achieved by selecting a random frame from the first 2 seconds of the recording being trained on (as all recordings started with a of couple seconds of rest before the first video started playing).

General hyperparameter ranges were found while experimenting with the dataset and used for all experiments, with no individual hyperparameter tuning for each model due to computational power limitations. This also avoided hyperparameter overfitting. Models were trained for 50 epochs at half-precision, with an AdamW optimizer with $1 \times 10^{-4}$ learning rate ($3 \times 10^{-5}$ for the multi-participant models) and $1 \times 10^{-3}$ weight decay ($\beta = 0.99$). Root mean square error was used as the loss function. Models were initialized with pretrained weights trained on the ImageNet-25k. A learning rate scheduler exponentially increased the learning rate from 0 to the target learning rate over the first 2 epochs, then decreased it linearly by a factor of 0.01 over the 50 training epochs.

## Online virtual reality control

To evaluate the final models in a realistic setting, we conducted a follow-up online control session for eight participants who took part in the first session and eight new participants. These sessions were conducted several months after the data collection (Table 1). For both groups, sessions did not include retraining or weight fine-tuning of the models. This allowed us to explore real-world performance in a setting comparable to what is expected from an ideal system and fairly evaluate model robustness after long periods of time.

Participants were positioned, standing, in front of an empty table while wearing a virtual reality headset (Vive Pro, Valve, Washington, United States) with the US bracelet positioned on their right arm. In the center of the table, a Leap Motion device (Leap Motion, Ultraleap, San Francisco, United States), a portable markerless optical motion capture system, was positioned. The Leap Motion was used to control the forearm position in 3D space, while the US system controlled the wrist and hand DoFs. Leap Motion data was also used to track the wrist and hand kinematics, so the predictions of both systems could be compared.

The wrist and hand 3D marker locations provided by the Leap Motion were converted into joint angles using the same inverse kinematics model used for marker-based data from the offline sessions. The only DoF where the inverse kinematics model could not be used was pronation-supination, as the the Leap Motion system was not able to track the ulnar and radial epicondyles. Instead, we directly used the arm rotation feature provided by the Leap Motion. This measurement was prone to angle wrapping and required manual correction by an expert.

In virtual reality, participants were presented a 3D environment with a rendering of an empty table (in the same place as the real one) and a floating hand model (Fig. 5a). The environment was developed with the Unity engine and did not use its default physics solver, but a realistic contact physics simulation provided by MuJoCo[79]. We adapted an existing MuJoCo model of the Modular Prosthetic Limb developed for the DARPA Haptix project as our virtual end effector[48,80]. The hand had 22 DoFs, comprised of three wrist, four thumb, and 15 finger hinge joints. The index, ring, and little finger supported abduction in addition to their three flexion joints. The metacarpal and interphalangeal DoFs for the non-thumb fingers were coupled, and the ring finger abduction was left passive, leading to 13 actuated DoFs. We coupled the control of each finger to form a single continuous grasp action. This resulted in a 4-dimensional control space, driving the wrist and grasp. We scaled the control signals from a [−1, 1] range to the corresponding joint ranges linearly, which were then applied as target angles in a proportional position control scheme. The passive joint damping of the model ensured stable convergence to the equilibrium pose. The kinematics of the subject's forearm provided by the Leap Motion system were used for the spatial configuration of the virtual hand within the environment, allowing subjects to intuitively position and orient it for tasks. However, none of the wrist or hand kinematics from the Leap Motion device was used to control the actuated DoFs, which relied solely on the US control model.

The virtual hand control task was performed to provide a more challenging evaluation of the stability and accuracy of the control. Unlike Unity's default 'gluing' solution that relies on external forces to maintain grasps for interacting with virtual objects, properly holding and manipulating virtual objects using contact physics requires precise control over the activation of each DoF.

The developed deep-learning models trained with the offline dataset were set to inference mode, and, in real time, provided predictions for each of the four wrist and hand DoFs that were transferred to the virtual hand. OBS Studio was used to record the participant's view of the VR environment, and a webcam showing the participant's hand moving in the real world. The VR environment, deep-learning model, and recording software all ran in a single workstation (GeForce RTX 3070).

With the model in inference mode providing predictions to the virtual hand, participants were loosely instructed to start moving their wrist and hand DoFs and given around 5 minutes to situate themselves and get accustomed to the control. When ready, participants were then presented, in sequence, five sets of virtual objects and guided in performing simple tasks by interacting with them. Table 2 describes the tasks. Tasks were loosely defined, and no strict start and end stages were enforced. Participants were allowed multiple tries in order to complete the tasks satisfactorily. It was observed that most of the challenge in completing the tasks was related to the limitations and idiosyncrasies of the contact physics simulation that made it difficult to perform very complex interactions.

## Metrics

To evaluate the model's offline performance, two metrics were calculated. All metrics were calculated per recording. For each model, the best performing epoch was used. First, the out-of-sample coefficient of

determination ($R^2$) was calculated. $R^2$ is a goodness of fit metric that ranges from $-\infty$ to 1, with zero representing a model that only outputs the mean value of the target variable. Second, the root mean square error (RMSE) was calculated. Before RMSE calculation, the normalized ground truth and predicted angles were converted back to degrees based on the range of motion measured for each recording session. Therefore, the RMSE had units of degrees. RMSE can range from 0 to $\infty$, with 0 representing a perfect prediction. Throughout the text the median instead of the mean was reported as, for both metrics, a small number of outliers can have significant impact on the mean.

For the online evaluation, the Pearson correlation coefficient (CC) was calculated between the Leap Motion angle predictions and the US predictions. Values were converted from angles to activation levels (based on the participant's range of motion recorded in the offline session) for the virtual hand before comparison. CC was used instead of $R^2$ as differences between pipelines caused variable baselines and scaling differences between predictions, causing $R^2$ to be mostly dominated by these variations and not representative of overall agreement between predictions.

## Statistics

In order to compare the distribution of performance metrics acquired from different models, a Wilcoxon Signed-Rank test was used. For each pairwise comparison, a two-tailed test was used to reject the null hypothesis (that both pairs come from the same distribution). If the null hypothesis was rejected, a single-tailed test was conducted to determine the pair with the greatest distribution. $p$-values reported come from the first two-tailed test. When comparing performance distributions for the multi-participant models, the same method was applied, but the Mann-Whitney U test was used since the samples were unpaired. The significance level was set as $p < 0.05$. Values lower than 0.0001 were written as $p < 0.0001$. In the case of multiple pairwise comparisons, the Bonferroni correction was applied to the significance level. For clarity in the figures, statistical significance values were not included, with numerical results aggregated in the Supplementary Discussions 2 and 3.

## Reporting summary

Further information on research design is available in the Nature Portfolio Reporting Summary linked to this article.

## Data availability

All processed data generated in this study and shown in both main text and supplementary material have been deposited on GitHub (github.com/BrunoSgambato/VR-HMI-US/paper) under a CC BY-NC 4.0 license. The raw ultrasound, motion capture data, and videos are not available due to its large size. The processed ultrasound and motion capture data for one volunteer participant has been deposited on Zenodo (https://doi.org/10.5281/zenodo.17296286) under CC BY-NC 4.0 license.

## Code availability

Code for generating all figures on the main text and supplementary materials is available on GitHub (github.com/BrunoSgambato/VR-HMI-US/paper). Custom code made for this study implementing the data pipeline, cross-validation, processing, augmentation, referencing strategy, and evaluation, is available at GitHub (github.com/BrunoSgambato/VR-HMI-US). The codebase also includes guidance and example configuration files that can be used with the dataset released on Zenodo (https://doi.org/10.5281/zenodo.17296286). All code is released under a CC BY-NC 4.0 license.

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

## Acknowledgements

We would like to thank Dr. Matthew S. Banger and Prof. Alison H. McGregor for the support with Motion Capture Acquisition. We acknowledge computational resources and support provided by the Imperial College Research Computing Service (https://doi.org/10.14469/hpc/2232). This work was supported by the European Union's Horizon 2020 Research and Innovation Programme through SOMA Project under Grant 899822 (D.F.). This work was supported by the EPSRC (Engineering and Physical Sciences Research Council), The VIVO Hub for Enhanced Independent Living grant (D.F.).

## Author contributions

B.G.S., B.K.H., D.Y.B., X.Y., M.X.T. and D.F. conceptualized the study. B.G.S., A.J. and M.F. developed the ultrasound Hardware and software. B.G.S. and B.K.H. developed the virtual reality models and validation. B.G.S. performed experimental data collection, model development and training, and analysis. B.G.S. and D.F. prepared the first draft of the manuscript. All authors edited the manuscript for important scientific content and all approved the final version.

## Competing interests

The authors declare no competing interests.
