## [Transparent Peer Review file · Nature Communications]

Virtual reality interactions via a user-generic ultrasound human-machine interface for wrist and hand tracking

Corresponding Author: Professor Dario Farina

Version 0:

Reviewer comments:

Reviewer #1

(Remarks to the Author)

This paper describes the design and evaluation of ultrasonic bracelet device worn on the arm to detect wrist and finger movements. The device uses 32 1MHz transducers arranged into two rings of 16. Each transducer transmits in turn and then all transducers, including the transmitter, operate as receivers in an ultrasound computer tomography set up operating at around 11Hz. The multi-receiver tomographic image data is then logged from 10 participants for processing by a neural network classifier. The first study compares the model's single-session, cross-session, and cross-participant performance, and a second study compares performance to an off-the-shelf LeapMotion optical VR hand tracker.

The paper is operating in a dense field of work, with many researchers actively working on novel signals and signal processing classifier pipelines to detect hand and wrist gestures and movements on the arm in a wearable form factor. A number of researchers have already published attempts to use ultrasound to sense hand movements, including key publications that demonstrate high accuracy in 2D (B-mode) imaging with wet electrodes, reasonable performance with 1D (A-mode) approaches, and strong results in prosthetic control, as cited in the paper.

The paper's introduction describes its niche as "more realistic use cases with participants being able to freely move in the environment", but this is undermined by the second study for which the Materials section makes it clear that positional data is being derived from the fixed optical system. The object grasping examples demonstrated in the accompanying video rely on both positional and rotational changes to support dextrous actions, and these could not be completed without the combination of optical and ultrasonic signals. The improvements in rotational sensing over the optical system cannot be used to replace the Leap Motion in the study, only to augment it to enhance rotational precision in the axis which the LeapMotion is least accurate. The ultrasound system would need to continue to be tied to a stationary exocentric optical viewpoint, which undermines the suggestion that limited work is required to achieve a fully portable and wearable system.

The conclusion section of the paper suggests that the key contribution is that this work presents the first design which has been demonstrated to be capable of robust cross-participant predictions. This reflects the results in Figure 4 which show that for pronation-supination and hand opening-closing, as the number of participants increases the cross-participant accuracy starts to approach the lower error bound of cross-session accuracy. However, cross-participant comparison to the literature for example to reference [32] is not fair, because a gesture set of 10 relatively similar movements is classified in that work, a much more challenging task. Further work with a fully wearable armband that isn't cited in the paper by Iravantchi et al* also performed similarly over a larger gesture set. While these works didn't go on to measure a high cross-participant accuracy, that is a function of the challenge of the large gesture set as much as it is performance of the hardware design or classifier, so it isn't clear that this work is a significant advance over the state of the art in this respect. Finally, the analysis in the paper used the first 70% of all recordings as training data and the last 30% as testing data. This raises a concern that the test data benefits from being scoped by the training process. A leave-one-out N-fold cross validation across training and testing data would be more robust to learning effects, and methodologically this reduces robustness in comparison of cross-participant performance with related works. Finally, the paper makes a claim that the cross-participant classifier model did not undergo fine-tuning. However, the signals were significantly curated through bespoke pre-processing augmentation, and the supplementary materials make it clear that the authors empirically experimented with a range of modern convolutional networks to identify a strong result, so there was significant optimisation of the pipeline independent of tuning the model itself. All this context needs to be developed in the round against existing literature to scope the contribution, and limits the

clarity of the improvement over the state of the art.

The Materials section gives much detail on how to reproduce the models, experimental designs and the data analysis. However, the bespoke 1MHz transducer design and the driving circuit/electronics aren't sufficiently detailed to enable reproducibility of the hardware aspects of the system.

* Interferi: Gesture sensing using on-body acoustic interferometry, Iravantchi Y., Zhang Y., Bernitsas E., Goel M. and Harrison C., in CHI 2019 Proceedings of the 2019 CHI Conference on Human Factors in Computing Systems

(Remarks on code availability)

Reviewer #2

(Remarks to the Author)

Title:

- I suggest to change the title from "generic" to "user generic", to make it more clear for the readers

Introduction:

- the introduction mentions how muscle fatigue is a challenge for using sEMG in kinematics. The same challenge should be applicable to US-based interfaces though

Results:

- Silicone elastomer: what is its duration? how many hours of use does it support?
- Dataset: will it be open sourced? and the code?
- page 5, line 144: "the echoes received by the non-active transducers have a variable spatial relationship with each other" I suggest to include a figure that shows the signal recorded by the other 31 channels when one channel is used for transmitting. How much of its signal is received by the other most distant transducer?
- page 6, line 165: reference missing
- Fig. 4 only reports R2 values. Please also include RMSE values.
- Fig. 4: can the authors comment further on the cases where R2 is mostly negative? a more detailed justification on the measured values would be valuable for the readers

Materials:

- Page 13, line 377: why did the authors chose to implement a single transmit/multiple receive strategy? why not emitting with all channels concurrently?
- Page 13, line 381: "The MoUSE acquires a second frame as soon as the first frame is transmitted to the connected laptop, therefore frame rate is variable depending on data transmission speed." How are data transmitted to the laptop? If I'm not mistaken the MoUsE uses an USB connection. What factors can influence the data transmission speed? What typical ranges of transmission speed did the authors experience? Please quantify the variability
- Page 13, line 383: the frame rate is 11 Hz. With 32 transducer, it means 352 Hz transmit pulse repetition frequency. I suggest to also mention this number explicitly in the paper for the sake of completeness
- please provide more details about the synchronization between the US platform and the Vicon (hardware connections?)
- follow up from the above question, Page 13, line 381: "At the start of US recording the Arduino generated a synchronization trigger recorded by the motion capture system." Is this the only signal used for synchronization? Is there any variability in the synchronization over time? Can the authors please comment and provide extra details on the accuracy of the synchronization?
- follow up question: What does "start" mean in this context? Is the start considered as the first recording or the start of each video instruction displayed to the participant? how is the syncornization between the displayed video instructions and the US acquisition?
- Page 14, line 394. Where does the image formation process occur? What is the time needed to create one image from A-mode scans?
- Page 15, line 489: "Before being used for model training angles were normalized by the minimum and maximum values of each recording session. Therefore the labels used were not raw angle values but a normalized DoF activation value." Several related works on EMG-based kinematics and US reported RMSE and MAE values to assess the results, with non-normalized labels. What happens if the labels are not normalized? It'd be useful for the reader if a comparative analysis of the impact of angle normalization vs no-normalization is included (e.g. in the supplementary materials)
- Page 16, line 528: "we selected the ConvNeXt Tiny as the most appropriate". Why is it the most appropriate?
- Page 16, line 540: typo "tunning"  "tuning"
- Page 16, line 546: Is there an estimate on the latency? What is the average time needed to complete each task?
- Page 17, line 605: "The RMSE represents the average error in units of degree at each point in time and can range from ∞ to 0, with 0 representing a perfect predictor model." If training angles are normalized, predictions will be on a normalized scale as well. Is any inverse computation performed to obtain predictions in absolute values?

(Remarks on code availability)

Reviewer #3

(Remarks to the Author)

(Remarks on code availability)

Version 1:

Reviewer comments:

Reviewer #1

(Remarks to the Author)

The authors provide helpful clarifications on the the stated objective of free movement of participants. Their intention was to contrast with highly restrictive posture constraints of the trunk and arm in prior work on ultrasound sensing of muscle activation, rather than aiming to support free movement across both body and the environment. This then allows for some integration of fixed optical tracking with the ultrasound system in order to demonstrate comparable performance to an all-optical approach. Essentially, the experimental set up is more focused on the reliability of the experimental design than the validity of the work against the proposed VR use case. The authors suggest elbow tracking may also be possible in future work from ultrasound data within the context of inverse kinematic constraint prediction. These arguments are convincing on their own merits, but they do suggest that, for fixed environments with relatively unconstrained movement, an ultrasound approach will struggle to demonstrate value in comparison to an optical system where multiple cameras can avoid typical issues such as occlusion, while fully unconstrained movement that could be enabled by an ultrasound system is beyond the capability reported in the paper.

The revisions which tighten up the precise scientific contributions of the work are positive. The authors have discriminated their work predicting multiple simultaneous targets from the existing literature which classifies larger discrete gesture sets. The key stand-out feature is the set of results which demonstrate independence and robustness in new participants. Again, this result doesn't place the approach in the same robustness bracket as optical VR approaches, but scoped according to ultrasound systems only is a clear advance.

The additions provided to enhance reproducibility of the work are appreciated, and through both the references and the additional characterisation, it will be possible to replicate the experimental setup described.

Overall, the paper now presents a stronger and clearer contribution to the field of ultrasonic gesture recognition. My remaining reservations relate to the proposed VR use case and the validity of the experimental studies as indicative of usage in this setting. It is quite clear that, at least for the capabilities demonstrated in this system, there will be no revolution in favour of ultrasound and away from optical approaches, and the approach still feels limited in production terms compared to existing robust hardware using EMG, visible or infrared signals. The work is provides a contribution to state of the art for ultrasonic gesture classification, but much less so for future VR approaches. For this reason the contribution may be too limited to warrant publication here.

(Remarks on code availability)

Reviewer #2

(Remarks to the Author)

The authors addressed all the previous comments. I only have one additional request for clarification related to angle normalization.

The authors mention that normalization is performed on a per-session basis, normalizing the angles by the max value measured during that session. While this approach is clear for training, it's not clear how this works in testing on unseen data.

Since the model works on normalized data, once the user starts to collect new data, what value is used for normalizing those input signals for the network? would it be the max value of the previous (known) session for that same user? A calibration session is first needed to evaluate what is the max range (for each DoF), as this is needed for the calibration?

(Remarks on code availability)

Reviewer #3

(Remarks to the Author)

(Remarks on code availability)

(Reply to Reviewers)

Virtual Reality Interactions via a User-Generic Ultrasound Human-Machine Interface for Wrist and Hand Tracking

Bruno Grandi Sgambato; Bálint K. Hodossy; Deren Yusuf Barsakcioglu; Xingchen Yang;
Anette Jakob; Marc Fournelle; Meng-Xing Tang; Dario Farina

We would like to express our appreciation and gratitude to the Reviewers for the time spent on reviewing the manuscript. The comments and suggestions were both useful and constructive, and have been taken into account in the revised version of the manuscript to improve its quality. A list of all changes is provided below, with an explanation and justification for each modification and point-to-point answers to each comment. Revisions are in red, blue and orange in the updated version of the manuscript to reply respectively to Reviewer 1, Reviewer 2 and both Reviewers simultaneously. Reviewer 3's questions and comments were included in the list of one of the other reviewers and answered together. During the process of reviewing the work and performing the changes suggested, a number of other small changes were made to improve clarity of sentences and correct small mistakes. While all major changes were highlighted, some very minor ones were not, for clarity. For lines numbering, please refer to the numbers in the manuscript. Before presenting detailed point-by-point answers to the reviewers, we highlight here a list of the major changes made, which include substantial revisions to the text, new analyses, and new experiments (included in both the main text and supplementary material):

- Major rewriting of the introduction with special focus on clarifying the challenges and accomplishments of the literature, our goals with this manuscript, and the main novelties presented.
- Expansion of the Discussion section, focusing on a more detailed overview of the closest literature comparisons and the challenges associated with US predictions in different settings.
- Preparation of an open-source repository implementing the main processing methods, augmentations, and other training strategies proposed in the manuscript. We also release a pre-processed example dataset for the community, including around 100,000 samples for one participant.
- Seven extra supplementary figures with examples of predictions to illustrate most of the cross-validations and model comparisons made on the main text.
- Two extra short experimental validations, included in the supplementary materials, quantifying the US frame rate, processing time, and latency.
- New experimental evaluation on the usage of angle normalization and its effect on prediction across single participant models.
- Minor changes on the text and figures to fix small mistakes and improve overall flow, and clarity.

We believe the extensive revisions made, which addressed all comments provided, have substantially improved the quality and clarity of the manuscript, and we hope the work will now better fit the requirements of the Journal.

Reviewer 1

This paper describes the design and evaluation of an ultrasonic bracelet device worn on the arm to detect wrist and finger movements. The device uses 32 1MHz transducers arranged into two rings of 16. Each transducer transmits in turn and then all transducers, including the transmitter, operate as receivers in an ultrasound computer tomography set up operating at around 11Hz. The multi-receiver tomographic image data is then logged from 10 participants for processing by a neural network classifier. The first study compares the model's

single-session, cross-session, and cross-participant performance, and a second study compares performance to an off-the-shelf LeapMotion optical VR hand tracker.

The paper is operating in a dense field of work, with many researchers actively working on novel signals and signal processing classifier pipelines to detect hand and wrist gestures and movements on the arm in a wearable form factor. A number of researchers have already published attempts to use ultrasound to sense hand movements, including key publications that demonstrate high accuracy in 2D (B-mode) imaging with wet electrodes, reasonable performance with 1D (A-mode) approaches, and strong results in prosthetic control, as cited in the paper.

Comment 1.1. The paper’s introduction describes its niche as ”more realistic use cases with participants being able to freely move in the environment”, but this is undermined by the second study for which the Materials section makes it clear that positional data is being derived from the fixed optical system. The object grasping examples demonstrated in the accompanying video rely on both positional and rotational changes to support dextrous actions, and these could not be completed without the combination of optical and ultrasonic signals. The improvements in rotational sensing over the optical system cannot be used to replace the Leap Motion in the study, only to augment it to enhance rotational precision in the axis which the LeapMotion is least accurate. The ultrasound system would need to continue to be tied to a stationary exocentric optical viewpoint, which undermines the suggestion that limited work is required to achieve a fully portable and wearable system.

Answer 1.1. We thank the reviewer for pointing out this matter, allowing us to provide a clarification. We have modified the manuscript text to better clarify our main goals, contributions and clarify how we envision the extension to a standalone US interface.

By referring to ”more realistic use cases with participants freely moving in the environment”, we intended to contrast our approach with previous US studies that relied on highly restrictive settings in order to achieve strong control results. We wanted to emphasize that allowing movement of the upper arm and trunk while still demonstrating strong predictive performance is both a critical feature of any realistic model and a significant challenge. In most previous studies, participants’ postures were constrained. We did not impose such constraints, not to suggest a wearable system enabling free movement, but rather to show that variations in arm and body posture can still yield strong results with our method, results compatible with the use of the interface under non-ideal laboratory conditions. Free movement itself is not essential for an interface; what is critical for a muscle interface is that different body and arm/hand postures typically alter muscle activation in ways that invalidate predictions. Addressing this challenge is precisely what we intended when referring to movement of the subject. To make this clearer, we have rewritten the paragraph to include the aforementioned concepts and better articulate our goals.

We strongly believe that a realistic system needs to allow participants to freely move their arms and body and also only require a single initial training session, with no lengthy recalibration for every donning and doffing event. Ideally, the control interface should simply allow participants to use the system immediately after wearing it for the first time, without any training. Our manuscript, therefore, primarily focused on exploring how these issues could be tackled and aimed at exploring what are the key methods needed to achieve this ideal system (answer 1.2 goes more in detail on what has been accomplished by the literature and how our work goes well beyond the state-of-the-art in these topics).

Regarding our real-time experiment, we want to highlight that its main goal was to provide evidence that the pipeline can be implemented in real-time and that the offline results translate to real-time scenarios (as in some cases strong offline results may not directly imply strong real-time performance [1]). We also aimed at demonstrating that control predictions were comparable to those of an optical solution, and that realistic challenging tasks could be accomplished.

We fully acknowledge that our experiment required the optical tracking of the elbow. As pointed out, the current implementation ties the US predictions to an exocentric frame of reference based on the leap motion readings for the elbow x,y,z position. It is possible however to still achieve some level of control and task completion without the leap motion, by fixing the elbow position to an egocentric frame of reference based on the headset FOV. With the current implementation this control would however feel constrictive and weird as participants would not have an ”elbow” movement. The extension of this implementation we propose, as future work, involves the addition of predictions for the elbow joint from US data as well (possibly also shoulder joint angles). In that situation, the shoulder position would be the one egocentrically fixed while the elbow, hand and finger joints positions in 3D space would be tied to the shoulder by a kinematic chain. In a smaller scale, the exact same approach is what is currently being used to control the fingers for hand opening and closing (the wrist/elbow global position is dependent on the optical reference

while the fingers x,y,z position is derived from the hand joints kinematic train). Coincidentally, a very recent work by Tang et al., published after our initial submission, explores the usage of US patches for prediction of shoulder, elbow and wrist to control a 4-DoF robotic manipulator in a virtual environment [2]. While the control used was simple and tailored to individual sessions, the work demonstrates very well the concept of controlling a complete "Arm-like" model through the simultaneous manipulation of angles between individual rigid segments using only US signals. Implementing this approach in our prototype could be achieved with one bracelet for the lower and one for the upper arm. While we are confident that the elbow joint flexion-extension angle can be robustly predicted, whether robust predictions of shoulder joint angles are possible from upper arm morphological deformations is an interesting but open question we wish to explore in future work.

Lastly, fixing elbow movement to the Leap Motion had a practical component to it. It forced participants to keep their hand unobstructed inside the FOV of the Leap Motion system and therefore allowed us to fairly compare predictions from both systems. Without this constraint participants would be able to move their hands outside the Leap Motion FOV (or position them in ways that the Leap Motion would struggle to detect) and comparison between the systems would not be fair. In our setup we can say that the Optical system was positioned to perform well and therefore allow fair comparisons.

Manuscript Changes: The introduction section has been updated to better explain our goals and how they are positioned against the current literature (Pages 2-3, Lines 85-93). The end of the discussion section going over limitations and future research directions has also been changed to include a longer discussion over the optical system dependency (Page 13, Lines 344-352).

Comment 1.2. The conclusion section of the paper suggests that the key contribution is that this work presents the first design which has been demonstrated to be capable of robust cross-participant predictions. This reflects the results in Figure 4 which show that for pronation-supination and hand opening-closing, as the number of participants increases the cross-participant accuracy starts to approach the lower error bound of cross-session accuracy. However, cross-participant comparison to the literature for example to reference [32] is not fair, because a gesture set of 10 relatively similar movements is classified in that work, a much more challenging task. Further work with a fully wearable armband that isn't cited in the paper by Iravantchi et al* also performed similarly over a larger gesture set. While these works didn't go on to measure a high cross-participant accuracy, that is a function of the challenge of the large gesture set as much as it is performance of the hardware design or classifier, so it isn't clear that this work is a significant advance over the state of the art in this respect.

* Interferi: Gesture sensing using on-body acoustic interferometry, Iravantchi Y., Zhang Y., Bernitsas E., Goel M. and Harrison C., in CHI 2019 Proceedings of the 2019 CHI Conference on Human Factors in Computing Systems

Answer 1.2. We thank the reviewer for the comment, which allowed us to better contextualize the state of US research and our novelty. We believe our descriptions of what has been accomplished in the literature and how challenging these tasks are was lacking and led to weak presentation of the novelty of our contribution. We have changed a portion of the introduction to better contextualize the challenges faced in the field and how our proposed methods can help the field advance. We have also expanded the discussion section when we compare our results with prior work. The discussion now clearly describes what has been accomplished in the most relevant related studies and how our approach/results advance the state-of-the-art. Lastly, we thank the reviewer for the new reference provided, we were not aware of that work and have now cited it in our discussion and future work section.

First, we include a longer introduction and discussion of the challenges of prediction of kinematics/kinetics over non-constrained settings. From the systematic review of the literature that we made, we have found around 120 published works on US used for interfacing since the first example in 2006. Out of them, only 8 studies have allowed mild upper arm, elbow or shoulder movement not directly related to the DoFs being controlled and only 3 allowed for more general movements by having participants performing actual functional system usage (all three cases related to prosthesis control).

The great majority of the studies described tests in settings that eliminate, or at least heavily minimize, any motion extraneous to the movements being decoded. Since US control features are derived from tissue motion, the classification/regression problem is heavily simplified in this way, as models need only to detect signal changes and associate them with commands based on their location. Akhlaghi et al. [3] and Yang et al. [4], for example, have both explored the limb position issue in B-mode and A-mode, respectively. Akhlaghi et al. reported an approximate 10% decrease in cross-validation accuracy (for a small set of 4

gestures) when testing across eight either seen or unseen limb positions. For variations on wrist position (forearm pronation and supination), however, Akhlaghi et al. identified substantial muscular deformation that even prevented the system from recognizing the motions at all if only trained in the neutral position, requiring specific training on each wrist position. Yang et al. found similar results for A-mode, with up to 12% reduction in performance (for 9 gestures) when testing during small variations in elbow and shoulder position (Shangguan et al. [5] also showed similar results with medium losses between 20-30% for elbow movements and up to 60% loss for rotations). These results demonstrate that loss of performance due to extraneous movements affect both A-mode and B-mode setups and can have similar effects regardless of the number of gestures or similarity between them. It also shows that movements that cause large muscular deformation, while intuitively simpler to decode, can have a disproportionate negative effect on model performance as models may struggle to compensate for them.

Therefore, we argue that setups like ours, which involve predicting multiple targets simultaneously, are at least as challenging as those with a larger number of discrete, non-simultaneous gestures. We also observe that while movements involving large muscle deformations (e.g., pronation-supination, flexion-extension, hand opening and closing) may be easier to track individually, they become significantly harder to track when occurring simultaneously. Our setup takes this to the extreme by also including dynamic movements of the arm happening simultaneously to the movements of the four DoFs being controlled, making the problem even more complicated.

Regarding approaches to cross-validation, we have found only 13 papers that evaluated cross-validation under scenarios involving sensor movement (our *Rotation* and *Position* tests), cross-session (our *Session* tests) or cross-participant situations. We have now included the most relevant references in our discussion, focusing on exploring the challenges faced by generalization works and describing what has been accomplished so far.

For the *Rotation* and *Position* groups, no truly comparable works have been found. We have however highlighted Kamatham et al.'s [6] work, which explores extracting different sets of shifted scanlines from B-mode images, with the results showing that, in some cases, for even very small shifts of 2.4 mm, performance could drop from an R^2 above 0.95 to below 0.2. Vostrikov et al. [7], on the other hand, explicitly mentioned that during cross-session tests bracelets were repositioned with some variation in rotation and position ($\approx 30^\circ$ and ≈ 3 cm) but did not explicitly attempt any systematic evaluation of these variations. Nonetheless, their final model reported achieving a 96% gesture accuracy in a simpler 4-class classification problem, providing evidence of the potential for generalizable models. Our work takes this a step further by providing a systematic evaluation of performance variations under large rotational and positional shifts.

Cross-session tests comprised the majority of the 13 works. In B-mode studies, [8] and McIntosh et al. [9] proposed that while models can be built that work well across-sessions, even slight shifts in probe placement across session can cause performance drops (with McIntosh et al. proposing optical flow corrections to reduce this impact). Yang et al. [4] and Shangguan et al. [5] identified this problem and proposed that performance across sessions could be maintained if systems included methods to guide participants into repositioning sensors correctly. They showed that classification accuracy could be improved by 23% over 6 classes with B-mode US and by 30% over 10 classes for A-mode, respectively. Lykourinas et al. [10] also showed the impact of cross-session evaluations (with models losing upwards of 50% classification accuracy), but instead proposed that models could be recalibrated by using data from the new session and adapting models in an unsupervised manner, recovering an average of 25% accuracy. The only work that can be closely compared to ours is the one from Spacone et al. [11] due to similar prediction goals. Overall, Spacone et al.'s cross-session RMSE results were worse when compared to ours (10.4° vs 10.1° on the flexion-extension DoF, 9.4° vs 5.8° for radial-ulnar deviation, and 13.5° vs 8.5° for hand open-close). This highlights the advantages of our approach, as our recording setting included more challenging free arm movement and a fourth DoF (pronation-supination).

Lastly, as far as we are aware, no other work has developed a US-based model capable of cross-participant generalization without model retraining. However, two works have explored cross-participant prediction by retraining models. First, Lian et al. [12] employed weight fine-tuning with an unspecified amount of data from new participants, and showed that performance could be fully recovered in most cases while computational time could be reduced. Similarly, Vostrikov et al. [7] used transfer learning, showing that while the classification head of a model needs to be trained for each participant, the feature extraction portion can be borrowed from other participants while maintaining performance. Our results point to a more generalized direction showing that cross-participant evaluations can be performed with US data - without any model adaptation. By combining models trained over multiple participants with the natural

ability of US to provide a neutral reference, we show that models are capable of robust performance in the previously unseen arm morphologies of new participants.

Manuscript Changes: The end of the introduction was changed to better introduce the problem we aimed at tackling, its importance for the field and better explain what we present (Page 3, lines 94-113). The discussion section was extensively changed to include more references to literature results and better descriptions of the challenges and novelties explored in our manuscript (Page 12, lines 275-315). Lastly, the provided reference was also included in the future work section in order to motivate future investigations involving interferometry approaches (Page 13, lines 370-372).

Comment 1.3. Finally, the analysis in the paper used the first 70% of all recordings as training data and the last 30% as testing data. This raises a concern that the test data benefits from being scoped by the training process. A leave-one-out N-fold cross validation across training and testing data would be more robust to learning effects, and methodologically this reduces robustness in comparison of cross-participant performance with related works.

Answer 1.3. Thank you for the comment. We believe our descriptions of the cross-validation groups and methods were confusing and led to some misunderstandings on how they were performed. We have changed the Data Collection section to better explain our cross-validation strategy.

To accomplish our goal of exploring the unique characteristics and issues behind various realistic intra- and cross-participant data shifts, we have defined six methods to split our dataset into training and testing groups. The first four groups were employed only for the single-participant models, the fifth group was used for both single- and multi-participant models, and the sixth group was only used for multi-participant models.

The first and easiest split, named *SameSet*, used the first 70% of samples from each recording (of one participant) as training while the testing set was formed by the last 30% of samples. It aimed at representing the simplest possible situation with all tested positions being present in the training data. This validation method has been the standard in US HMI literature (as discussed in answer 1.2), with some studies even shuffling the samples prior to splitting. As mentioned, this separation of data leads to concerns of information leakage between the training and testing datasets as US samples being tested are likely to hold reasonable similarities to the ones in the training set. Unlike other works, we ensure greater uniqueness in the test samples by only performing each movement once. Making even this simple case likely more challenging than cases normally found in the US literature. This was the only group where individual recordings were split between training and testing.

The 70%-30% split was exclusively used for the *SameSet* cross-validation. All other splits followed a standard leave-one-out cross-validation strategy, where only complete recordings were split between training and testing. This allowed us to simulate realistic usage situations and eliminate concerns about information leakage. For example, the *Rotation* group simulates a participant repositioning the bracelet to a different orientation than its original one after a short break. In this group, data was separated based on four pre-defined rotational positions, with three reserved for training and the last for testing, forming a standard leave-one-out 4-fold cross-validation setup.

Regarding our cross-participant results, we want to highlight that no data from the participants being tested was used to train the models. This represents a notable improvement over the current state of the art. To our knowledge, all existing US-based models that evaluate predictions across participants have included data from test subjects in the training process, even if only for model adaptation or fine-tuning [7, 12].

Manuscript Changes: We have changed the Data Collection section of the manuscript to better explain how we conducted cross-validation and how our methods separate us from the state-of-the-art (Page 16, Lines 474-485 and 495-503).

Comment 1.4. Finally, the paper makes a claim that the cross-participant classifier model did not undergo fine-tuning. However, the signals were significantly curated through bespoke pre-processing augmentation, and the supplementary materials make it clear that the authors empirically experimented with a range of modern convolutional networks to identify a strong result, so there was significant optimisation of the pipeline independent of tuning the model itself. All this context needs to be developed in the round against existing literature to scope the contribution, and limits the clarity of the improvement over the state of the art.

Answer 1.4. Thank you for mentioning this. We believe that we have failed to provide a clear explanation of what fine-tuning entails in the context of our work and may have thus caused some misunderstandings. We

have now made it clearer in the introduction, that in our work, fine-tuning refers to the approach of adapting pre-trained deep learning model weights to new tasks by using small amounts of data from the new situations.

In deep learning literature fine-tuning is sometimes used to refer to the process of further training a pre-trained model on a new task. In the context of HMI works, fine-tuning (sometimes referred to as model adaptation) normally refers to using a small amount of data to adapt a model to a new cross-validation situation, be at a new position of the sensors, a new recording session or a new participant.

The idea behind fine-tuning comes from the assumption that a model trained in one context (e.g., a group of participants) may not generalize to a new context (e.g., new participants). By using small amounts of data from the new situation to tune the model's weights, a balance can be reached between performance and practicality. From the US HMI literature, for example, Lykourinas et al. [10] proposed the use of unsupervised model adaptation to improve cross-session predictions. The work shows that, in their settings, cross-session evaluations caused classification accuracy drops close to 50%. However, by using unlabeled data from the new session the models could be adapted and recover up to 25% accuracy.

On the topic of cross-participant predictions, the only other works we are aware of that have explored this setting both used model adaptation methods. Lian et al. [12] employed weight fine-tuning by retraining models with data from new participants, showing that depending on what parts of their network were retrained, performance could be fully recovered in most cases. Vostrikov et al. [7], on the other hand, used transfer learning, meaning that a part of their model (the feature extraction network) was trained on a different participant, while only the classification head was fully trained on data from the target participant. The main advantages of both methods, when compared to fully training a model on each participant, are related to reduced data requirements and shorter training times.

While we could have employed fine-tuning for our methods, and likely reached better performance, we wanted to highlight the fairly surprising result that our US models were already capable of a high degree of accuracy without being specifically tuned to new participants. This is the first time this result is observed. While comparison is difficult, for sEMG interfaces this level of generalization seems to require considerably larger datasets, as evidenced, for example, from a recent publication by Meta in Nature [13]. This also seems to contradict the assumption that the higher resolution and sensitivity of US make it more prone to poor robustness. This is the central takeaway message and novelty we aimed to emphasize.

Regarding our exploration of processing and methods, it is true that multiple processing pipelines and models were explored to reach our results and that a naive implementation would not achieve the performance we have. Nonetheless, the pipeline was optimized towards a model capable of general performance and not specifically tuned to perform well in any single participant. This is demonstrated by the fact that during the online experiment, after the models had all been trained and locked in place, the model performed well in eight new participants, without any re-training. We believe that the processing guidelines we proposed are general enough to be applicable to other groups exploring A-mode US-based HMI, regardless of hardware, sensors, or prediction targets. Its also possible that some of the processing methods and augmentation approaches we proposed could be leveraged by B-mode US HMI studies.

Manuscript Changes: We have updated the introduction to clarify what fine-tuning refers to, in the context of our work (Page 3, lines 102-104). The paragraphs of the discussion have also been adapted to clarify what other authors have presented in terms of model re-training and how our approach differs from these previous solutions (Page 12, lines 297-315).

Comment 1.5. The Materials section gives much detail on how to reproduce the models, experimental designs and the data analysis. However, the bespoke 1MHz transducer design and the driving circuit/electronics aren't sufficiently detailed to enable reproducibility of the hardware aspects of the system.

Answer 1.5. We are happy to hear that our Materials and Methods section is up to standard and covers all relevant experimental design and data analysis decisions.

Regarding the electronics, a more complete description of it is available in a separate publication [14]. We have also added a reference to an older conference proceeding describing the system as it also includes some extra information and a block diagram of the system layout [15]. We believe these references should support reproducibility of the hardware aspects of the system.

Nonetheless, we wish to mention that our contributions in this paper are not tied to any specific hardware. The acquisition pipeline and processing we implemented could theoretically be implemented in any research-grade US system that allows for setting arbitrary transmit-receive patterns and for raw data to be recorded (e.g., Verasonics Vantage, us4us).

Regarding the 1 MHz transducers we realized that our final submission of the manuscript Supplementary Materials was missing a section including some characterization of the transducers. We have now added this section back in and also expanded it to cover a description of the manufacturing procedure of the transducers we used.

Manuscript Changes: A new reference to the hardware used was included in the text (Page 14, Lines 392-395). The Supplementary Discussion 1 was included and expanded to include details on the manufacturing process of the transducers used.

Comment 2.1. I suggest to change the title from "generic" to "user generic", to make it more clear for the readers

Answer 2.1. Thank you for the suggestion. We agree with the inclusion of "user" in the title to more clearly communicate our main results

Manuscript Changes: The title has been changed as suggested (Page 1, lines 1-2)

Comment 2.2. The introduction mentions how muscle fatigue is a challenge for using sEMG in kinematics. The same challenge should be applicable to US-based interfaces though

Answer 2.2. Thank you for pointing this out. We agree that without further clarification, this mention can be misunderstood. Interestingly, evidence from the literature points out that US approaches, when compared against sEMG systems, are more robust to changes due to fatigue. Work by Zeng et al. compared sEMG and A-mode US in a force tracking task under fatiguing conditions. sEMG alone outperformed US in force tracking but US was more robust against the fatigue condition [16]. This was further validated in a follow-up work via comparison of feature space repeatability and separability [17]. Moreover, it has been shown that US performance during fatigue conditions can be further improved via specific adaptive networks [18].

Manuscript Changes: We have clarified the general statement we made on the introduction about other US works. We specifically mention a study showing that US signals, when compared to sEMG, are more robust against fatigue (Page 2, lines 84-85).

Comment 2.3. Silicone elastomer: what is its duration? how many hours of use does it support?

Answer 2.3. Thank you for bringing this up. Unlike water-based coupling solutions (traditional US gel or hydrogel), silicone is an inert and stable material. Therefore, as it does not dry over time, it can allow for coupling over considerably longer periods of time. The main limiting factor for silicone-based coupling is simply related to normal wear and tear of the layers. Specifically, silicone slowly peels off from the transducer's surface due to continuous usage and mechanical stress. We have designed the bottom surface of the holder to have a number of small grooves, holes at the edge, and a small wall along the sides so that peeling off is greatly reduced. While we have not systematically tested the resistance and durability of the solution, we can confirm that we have used a single application of silicone for upwards of 4 months over dozens of experimental sessions with few visible changes to the surfaces and no visible changes to the signal quality.

Manuscript Changes: We have identified that a session describing this and other US characterization on the Supplementary Materials was missing from the submitted version. This has been added back (Supplementary Discussion 1). We have also added a comment on the durability of the silicone and referenced our discussion on the supplementary materials in the main text (Page 4, lines 121-122)

Comment 2.4. Dataset: will it be open sourced? and the code?

Answer 2.4. Thanks for mentioning this. Much of the source code developed is related to communicating with the acquisition hardware, generating the graphical interface and interfacing with the virtual environment. The final version of the codes for processing and training the networks is also highly integrated with specifics of our setups and not very portable.

Nonetheless, in the interest on supporting further research by other groups we have prepared a alternative version of the code developed, including the portions of the methods proposed that do not require specific hardware integration. Together with this manuscript we will release a open-source repository in GitHub to exemplify the methods and processing approaches we describe in the manuscript. The repository implements the exact same models and cross validation loops we ran for the manuscript, as well as modularized and easy to use versions of our proposed data augmentations and other training strategies.

Raw US data is, however, very large in size and challenging to share. Still, we have prepared a processed version (significant smaller in size) of the US files for one participant (a new participant, not included in the manuscript experiments). We will release this dataset on Zenodo in a open-source format and include specific configuration files to allow researchers to very easily run models under the same cross validation groups we used. The dataset will contain approximately 100,000 samples of paired processed US data

(following the processing steps described in the manuscript) and normalized joint labels. As far as we are aware, this is only the second ever open-source dataset on US-based HMI. We are happy to explore sharing the raw data, as reasonable, in a case-by-case basis.

The files that will be released on GitHub were included in a zipped format for reviewers reference.

Comment 2.5. page 5, line 144: "the echoes received by the non-active transducers have a variable spatial relationship with each other" I suggest to include a figure that shows the signal recorded by the other 31 channels when one channel is used for transmitting. How much of its signal is received by the other most distant transducer?

Answer 2.5. We have expanded Figure 1 in the manuscript to include an illustration of what the acquisition process looks like. The expanded figure also includes an example of a frame of the raw US data showing how the echoes received by each transducer during different transmissions look like, as requested. To answer your questions specifically, at each transmission event, the echoes received by the active transducer are typically the strongest, as expected. This is also true in some cases for the transducers closest to it (as visible on the middle graph of Figure 1D, with channel 16 transmitting, channels 14, 18, 20 also recorded large echoes). The signals in other channels, however, are highly dependent on the position of internal body structures and how the echoes are reflected (or transmitted) through them. Therefore, the farthest transducers are not always necessarily the ones with weaker echoes. Normally, some of the sensors are directly on top of a bone and thus typically are the ones that receive fewer echoes from transmissions generated by other sensors.

Manuscript Changes: We have expanded Figure 1 with an illustration of the acquisition method and an example of the raw data collected (Figure 1, Page 4).

Comment 2.6. page 6, line 165: reference missing

Answer 2.6. Thank you very much for noticing that. We have added the intended reference of a recent effort on an EMG dataset aiming to improve intra-session generalization.

Manuscript Changes: Reference was added (Page 6, line 165).

Comment 2.7. Fig. 4 only reports R2 values. Please also include RMSE values.

Answer 2.7. We have included a new image on the Supplementary Materials with the cross-session and cross-participant performances based on the RMSE measures as suggested.

Manuscript Changes: Figure S4 has been included in the Supplementary Materials with the RMSE results for the cross-session and cross-participant evaluations (Supplementary Figure S5).

Comment 2.8. Fig. 4: can the authors comment further on the cases where R2 is mostly negative? a more detailed justification on the measured values would be valuable for the readers

Answer 2.8. Thank you for the suggestion. Based on this comment, we realized that we ended up not including any examples of predictions for the offline sessions in our manuscript. This escaped our attention and we have now corrected this by including 7 new figures (each with two examples) in the Supplementary Materials covering predictions of different models in the diverse cross-validation setups we employed. We also included two highlighted examples on Figure 4 of the main text. In the Supplementary Materials, each figure legend also comes with a small commentary on some interesting observations in the examples showcased.

Figure S11 includes four single participant models trained on the different cross-validation strategies, predicting the same unseen recordings. It shows that prediction offset was one of the main causes for lower performance in some of the cross-validation situations. Figures S12, S13, S14 and S15 focus on comparing the performance of the three single participant models with each figure corresponding to a different cross-validation method. It aims to show real examples on how the introduction of both the convolutional models and the augmentation strategies enhances the predictive power of the system for each cross-validation group. Figure S16 includes results for the multi-participant models when tasked with cross-participant predictions and compares the performance of these models depending on the number of participants used on the training set. In it, we showcase the improvements to model generalizability when training on more participants. Lastly, Figure S17 showcases the cross-participant predictions on multi-participant models but focuses on the impact of our proposed referencing strategy and how it strongly improves prediction performance.

To directly answer the question asked, poor-performing examples can be mostly divided into two groups: Poor quality predictions or prediction offsets. Poor quality predictions can be seen on Pronation-Supination DoF of Figure S17 example A or the Flexion-Extension DoF of Figure S14 example A. In both, models struggled to predict anything and mostly predicted values around the mean. The R^2 for these cases tends to hover around 0 but is rarely very negative. In other situations the models are clearly able to follow the trends of the recordings but their predictions suffer from a noticeable offset from the ground truth targets. The hand Open-Close DoF of Figure S11 example B shows that even small offsets can have drastic effects in the R^2 . The hand Open-Close DoF of Figure S13 example B and the Flexion-Extension DoF of Figure S17 shows more drastic situations, where even if the predictions somewhat follow the trend of changes, the very large offsets cause the R^2 values to be close to -2 and -9, respectively.

Manuscript Changes: Figures S11-S17 were included in the Supplementary Materials showing examples of predictions made by various models in varied situations. References to them were added in the main text (Page 6, lines 151-152 and page 8, line 196). Two other example predictions were added to Figure 4 with to illustrate the importance of multi-participant training and referencing in achieving generalization and referenced in text (Page 8, lines 213-214).

Comment 2.9. Page 13, line 377: why did the authors chose to implement a single transmit/multiple receive strategy? why not emitting with all channels concurrently?

Answer 2.9. Thank you for your question. The standard approach for A-mode acquisition is the single-transmit, single-receive strategy where each transducer works completely independently of the others. In a previous work [19], we were inspired by non-destructive testing methods that employ Full-Matrix Capture and identified the possibility of recording extra information from the forearm responses to a single transmission using all other available transducers. In B-mode imaging, transmissions with multiple active transducers are widely used as they can, for example, allow for ultra fast frame rates. For A-mode we have briefly explored having multiple active transducers in a single event, but have not found many advantages in doing so. Transmission with multiple elements is normally performed when the spatial relationship between elements is known a priori, and it is therefore possible to create specific wavefronts. Due to the variable position between our transducers, this would not be possible in our case. Nonetheless, this is an interesting research direction and we believe there is potential for multi-transmission setups aiming to beamform the data from multiple sensors into a single tomographic image. We have also been made aware by Reviewer 1 of a literature reference that aimed at performing acoustic interferometry by using multiple transmissions from a bracelet of sensors and included this approach as another interesting possibility of leveraging multi-transmit strategies [20] in future works.

Manuscript Changes: We have included a comment of the potential for beamforming of bracelet data as a suggestion on future works (Page 12, lines 327-328).

Comment 2.10. Page 13, line 381: "The MoUSE acquires a second frame as soon as the first frame is transmitted to the connected laptop, therefore frame rate is variable depending on data transmission speed." How are data transmitted to the laptop? If I'm not mistaken the MoUSE uses an USB connection. What factors can influence the data transmission speed? What typical ranges of transmission speed did the authors experience? Please quantify the variability

Answer 2.10. Thank you for the question. Indeed the MoUSE system connects to the host computer via USB. We have included a new Supplementary discussion that goes over this process in more detail (Supplementary Discussion 2).

Briefly, the MoUSE uses a USB 3.0 interface. The system is designed in a way that after a single acquisition is defined, the MoUSE performs it, transfers the data from its internal memory to the host computer and repeats. Therefore, the amount of data acquired in a single acquisition is the main factor affecting the achievable frame rate. Using fewer transmit-receive combinations, or reducing recording time can therefore improve the frame-rate. Depending on the configuration, very fast frame rates (above 300 frames per second) can be acquired. The system is explained in more detail in a separate publication [14].

We conducted a new short experiment to properly quantify the frame-rate and its variability, with results shown in Figure S2 and discussed in the Supplementary Discussion 2. We performed a dummy virtual reality experiment (identical to the experiments but with no participant) lasting approximately 2 minutes and measured the time between frames throughout the session. Results showed that the system acquires at an average frame rate of 11.8 ± 0.5 Hz, with instantaneous frame rates varying between 7.8 Hz and 13.5 Hz.

Manuscript Changes: Changes were made into the Methods Section to include references to the new quantification results and comments (Page 14, lines 388-391). A new Section and figure were added to the Supplementary Materials (Supplementary Discussion 2 and Figure S2).

Comment 2.11. Page 13, line 383: the frame rate is 11 Hz. With 32 transducer, it means 352 Hz transmit pulse repetition frequency. I suggest to also mention this number explicitly in the paper for the sake of completeness

Answer 2.11. Thank you for the suggestion. We have included this suggestion in the manuscript. More comments on the specifics of our frame rate quantification were included in Answer 2.10.

Manuscript Changes: Information was included on the new Supplementary Discussion 2 section.

Comment 2.12. please provide more details about the synchronization between the US platform and the Vicon (hardware connections?)

Answer 2.12. Thank you for the question. Given the similarity of this comment and comment 2.13 we have provided a single combined answer to both bellow.

Manuscript Changes: Please refer to answer 2.13

Comment 2.13. follow up from the above question, Page 13, line 381: "At the start of US recording the Arduino generated a synchronization trigger recorded by the motion capture system." Is this the only signal used for synchronization? Is there any variability in the synchronization over time? Can the authors please comment and provide extra details on the accuracy of the synchronization?

Answer 2.13. Thank you for the question. We have expanded the paragraph in the Methods section describing how we synchronized both systems.

Both US and Vicon MOCAP data are paired with a timestamp array that contains the time between each frame and the start of the recording (in milliseconds). The start of the recording was determined both for the VICON and US systems by using an Arduino and the Vicon Lock Lab. When the US system started recording, the Arduino sent a pulse to the Start/Stop ports on the Lock Lab, triggering the start of the VICON recordings. In some cases, we also connected the trigger pulse sent by the Arduino to the Vicon analog port. Recordings were then initially aligned using either the start time or the trigger, when available. Recordings were then manually checked to ensure the start of the first movement (from rest) matched in both US and MOCAP data as some slight variation in the Vicon starting time was observed.

During model training, the DataLoader finds, for each US sample, the MOCAP frame with the closest timestamp and uses it as its label. The timestamp array for the US system was obtained during acquisition, as the custom recording interface for it saves the timestamps received with each acquisition. For the Vicon, as the system frame rate is very consistent, the timestamp array was generated from equally spaced samples. In our recordings the number of samples acquired by the Vicon, when compared to the expected number for the recording period, varied by 20.2 ± 22.9 frames. This means that, at 200 frames per second, timing of frames varied by an average of 100 ms during the full recordings (2 minutes for the wrist recordings and 1 minute for the functional recordings).

Manuscript Changes: The synchronization section the the manuscript was rewritten to more completely explain the synchronization procedure (Page 15, 412-418).

Comment 2.14. follow up question: What does "start" mean in this context? Is the start considered as the first recording or the start of each video instruction displayed to the participant? how is the syncornization between the displayed video instructions and the US acquisition?

Answer 2.14. Thank you for the question. In the sentence, by "start", we meant the start of US recording. Videos were manually started after a few seconds of the US/MOCAP recording starting. Therefore, there was a variable delay of a couple of seconds between the start of the recording and the start of the videos.

This was not a problem, as there was no synchronization between the videos being displayed and the recordings. The order of the small instruction videos was randomized before being sequentially played in each repetition, and the order was not recorded. This was completely intentional, as videos had the purposed of only guiding participants into performing varied motions. In this work we focused on more realistic settings cross-validation settings, so there was no need to record the order in which specific movements were performed. Movements were only performed once per bracelet position to avoid cross-validations that were to simple. As the labels used were generated purely by the MOCAP system, this had no effect on the recordings.

In reality, in multiple cases it was observed that participants were delayed relative to the videos or performed incorrect movements by mistake. Allowing imperfect imitation of videos improved the efficiency of data collection as all recordings could be used regardless of whether participants made small mistakes when imitating some. This and other small optimizations were key in enabling us to collect a large, varied dataset.

Manuscript Changes: An extra mention of the non-synchronization between videos and recordings as well as the freedom for participants to make mistakes when imitating the videos was included in the Data Collection section (Page 16, 477-479).

Comment 2.15. Page 14, line 394. Where does the image formation process occur? What is the time needed to create one image from A-mode scans?

Answer 2.15. Thank you for pointing this out. We conducted a proper quantification of this process and included it, along with some general comments, in a new section on the Supplementary Materials.

For the offline experiment, data was saved after each acquisition in its raw format and processing was done asynchronously in a high-performance cluster. For the online experiments, the same image formation pipeline was implemented to run in real time by the host computer. Model inference was performed on the host computer GPU, while all other processes were performed on its CPU. The pipeline implemented to run in real time was almost identical to the one used offline, with the only difference being that the downsampling step was split into two sub-steps with one performed just before the Hilbert enveloping. This was done to improve processing time and guarantee that a prediction would be ready before a new frame was available. Figure S2C-D shows the times we measured for key parts of the image formation pipeline. Similarly to what was done for frame-rate measurements, these values were acquired over a 2-minute dummy experiment. Overall processing took an average of 60.6 ± 8.5 ms per frame.

Manuscript Changes: Changes were made into the Methods Section to include references to the new quantification results and comments (Page 15, lines 415-418). A new Section and figure were added to the Supplementary Materials (Supplementary Discussion 2 and Figure S2.)

Comment 2.16. Page 15, line 489: "Before being used for model training angles were normalized by the minimum and maximum values of each recording session. Therefore the labels used were not raw angle values but a normalized DoF activation value." Several related works on EMG-based kinematics and US reported RMSE and MAE values to assess the results, with non-normalized labels. What happens if the labels are not normalized? It'd be useful for the reader if a comparative analysis of the impact of angle normalization vs no-normalization is included (e.g. in the supplementary materials)

Answer 2.16. Thank you for the question. The question on whether to directly predict joint angles or predict a normalized activation value is an interesting one that we have explored considerably. We have now included a new section on the Supplementary Materials that discusses this issue and provides some results on the advantages of applying it (Supplementary Discussion 9).

Our reasoning behind using normalized values was related to cross-session and cross-participant predictions as different people can have fairly variable ranges of motion. In our sessions we also saw that the range of motion measured varied between sessions, likely due to slight variations on marker positioning and different effort levels by each participant to reach their "maximum" activations.

Due to this we were worried that models would tend to focus on memorizing specific patterns for each session/participant instead of learning more general inferences about the movements being performed, limiting generalization to new sessions/participants. We therefore proposed a normalization approach where angles were normalized, per session, based on the range of motion recorded for it. To quantify whether this would be a good strategy we trained single-participant models on raw angles and on normalized values, and compared their performance.

The new Figure S11 compares the results obtained for models trained with joint angles or normalized values. Interestingly, even for these results obtained in single-participant models a significant performance difference was found for most cross-validation groups (all except the *Rotation* group). As expected the cross-session evaluation was the one that most benefited from the normalization, but the effects from it also seem to improve the model even if they have specifically been trained with examples from both sessions.

Lastly, we also chose the normalization approach for a very practical purpose. By using normalized angles we can guarantee that all participants can, ideally, have control over the complete "range of motion" of the

virtual hand. If using raw angle predictions and mapping those to the virtual hand some participants may be more limited in controlling the hand. As our intention with this work is to demonstrate the potential of US as a control interface, we have prioritized ease of control.

Manuscript Changes: A new section and figure were included in the Supplementary Materials showing a new experiment exploring the effect of angle normalization on prediction performance (Supplementary Discussion 9 and Supplementary Figure 11). A reference to this was added on the main text (Page 17, lines 569-571).

Comment 2.17. Page 16, line 528: "we selected the ConvNeXt Tiny as the most appropriate". Why is it the most appropriate?

Answer 2.17. Thank you for the question. We see that this sentence reads a bit confusing. We meant that the Tiny model size was the most appropriate for the work. We have changed the section to make it more clear how we went about choosing the model

First, we have experimented with a range of convolutional backbones (e.g., ResNet, DenseNet, ViTs, SwinT). Overall all models were able to learn reasonably well, but the ConvNeXts proved to be slightly ahead in terms of performance and easier to train. Regarding the sizes we explored models with different parameter counts by using the sizes and shapes proposed in the original paper (Tiny, Small, Base and Large). We have preliminary evaluated sizes by training single-participants models with each model size (only in a couple cross-validation methods) and found little to no change in model performance regardless of size. Its likely that an even more diverse dataset is needed to leverage larger models appropriately.

Manuscript Changes: We have changed the Models and Model Training section to reflect the the model backbone was chosen more clearly (Page 18, 600-605).

Comment 2.18. Page 16, line 540: typo "tunning" -> "tuning"

Answer 2.18. Thank you for spotting this issue. We have also carefully proofread the manuscript once again and found a couple of other errors and opportunities for better phrasing.

Manuscript Changes: The error was fixed on Page 16. A number of other small phrasing issues and grammatical mistakes were also spotted and fixed.

Comment 2.19. Page 16, line 546: Is there an estimate on the latency? What is the average time needed to complete each task?

Answer 2.19. Thank you for both questions.

Regarding the latency we can quantify it as the time between measurement and prediction. Based on the US workings we know that it takes an average of 85 ms (explored on answer 2.11) to transfer a new frame to the host computer and that the image formation and processing pipeline takes an average of 60 ms (explored on answer 2.15). Therefore, the latency from movement to prediction can be estimated at approximately 140 ms, on average. Research on prosthesis has pointed that delays, in that domain, should normally be kept lower than 100 ms [21]. Our application however behaves differently as control is continuous (instead of on and off) and therefore delays are less impactful, as the difference from subsequent predictions tends to be small. We therefore don't believe this had a significant impact in performance. During our online tests, the delay was difficult to notice (for example on the Supplementary Movie 1) and was not mentioned by any participant as a source of issues. The latency for movements was only clearly noticeable when performing very fast movements from rest. Lastly, the current latency is simply a limitation of the hardware we had available and our implementation. Usage of faster communication protocols for faster data transfer and porting all image processing to the GPU are two ways of dramatically improving latency, for example.

Regarding the average time to complete each task we have sadly not recorded this data. Overall each session lasted around 30 min. With the complete set of five tasks lasting somewhere between 10-25 min depending on the participant. Some participants were very quickly able to perform all five tasks with little training and trials while others struggled a bit more with the physics simulation and object interaction.

Manuscript Changes: We have include the information on the latency as part of the new Supplementary Discussion 2 that also discusses the US system frame-rate and the time taken by each processing step in the prediction pipeline.

Comment 2.20. Page 17, line 605: "The RMSE represents the average error in units of degree at each point in time and can range from ∞ to 0, with 0 representing a perfect predictor model." If training angles are normalized, predictions will be on a normalized scale as well. Is any inverse computation performed to obtain predictions in absolute values?

Answer 2.20. Thank you for bringing this up. We have indeed converted the normalized predictions back into angles before calculating the RMSE. This information was mistakenly omitted on the original manuscript, but we have now updated it to include this process. The same process was also used to show the predictions examples we have now included.

For each recording, the range of motion recorded for that individual participant, in that individual session, was used to scale the normalized activation values back to their original angular range. For the ground truth values, this yields the same values as the ones obtained by the inverse kinematics pipeline. For the predicted values, this could be considered improper as the range would theoretically not be known for new participants. However as we only use these values to provide comparisons to the ground truth and to calculate the RMSE in units of degrees (and not during real system control) we consider this approach reasonable.

As mentioned in Question 2.16 we are aware of the RMSE being a highly used metric in other studies and that it is normally reported in terms of degrees. Therefore we included this conversion back to angle values so our results could also be more easily compared to present and future literature results.

Manuscript Changes: Manuscript Material and Methods section was slightly altered to add this information (Page 16, Lines 500-5002 and Page 19, Lines 616-619).

Reviewer 3

Comment 3.1. I co-reviewed this manuscript with one of the reviewers who provided the listed reports. This is part of the Nature Communications initiative to facilitate training in peer review and to provide appropriate recognition for Early Career Researchers who co-review manuscripts.

Answer 3.1. We thank the reviewer for its efforts in thoroughly reviewing the manuscript. We hope to have answered your questions and addressed your concerns in the answers above.

References

1. Jiang N, Vujaklija I, Rehbaum H, Graimann B, and Farina D. Is Accurate Mapping of EMG Signals on Kinematics Needed for Precise Online Myoelectric Control? *IEEE Transactions on Neural Systems and Rehabilitation Engineering* 2014;22:549–58.
2. Tang Z, Wu Y, Qu M, et al. Synchronous Gesture Recognition and Arm Joint Angle Monitoring for Human-Machine Interaction Using Multiple Flexible Ultrasonic Patches. *Advanced Functional Materials* 2025;n/a:e05131.
3. Akhlaghi N, Baker CA, Lahlou M, et al. Real-Time Classification of Hand Motions Using Ultrasound Imaging of Forearm Muscles. *IEEE Transactions on Biomedical Engineering* 2016;63:1687–98.
4. Yang X, Zhou D, Zhou Y, Huang Y, and Liu H. Towards Zero Re-Training for Long-Term Hand Gesture Recognition via Ultrasound Sensing. *IEEE Journal of Biomedical and Health Informatics* 2019;23:1639–46.
5. Shangguan Q, Lian Y, Cai S, Wu J, Yao L, and Lu Z. DANN-Repositing Strategy for Zero Retraining Long-Term Hand Gesture Recognition Using Wearable A-Mode Ultrasound. *IEEE Transactions on Instrumentation and Measurement* 2024;73:1–11.
6. Kamatham AT, Alzamani M, Dockum A, Sikdar S, and Mukherjee B. SonoMyoNet: A Convolutional Neural Network for Predicting Isometric Force From Highly Sparse Ultrasound Images. *IEEE Transactions on Human-Machine Systems* 2024;54:317–24.
7. Vostrikov S, Anderegg M, Benini L, and Cossettini A. Unsupervised Feature Extraction From Raw Data for Gesture Recognition With Wearable Ultralow-Power Ultrasound. *IEEE Transactions on Ultrasonics, Ferroelectrics, and Frequency Control* 2024;71:831–41.
8. Khan AA, Dhawan A, Akhlaghi N, Majdi JA, and Sikdar S. Application of wavelet scattering networks in classification of ultrasound image sequences. In: *2017 IEEE International Ultrasonics Symposium (IUS)*. 2017:1–4. DOI: [10.1109/ULTSYM.2017.8091649](https://doi.org/10.1109/ULTSYM.2017.8091649).
9. McIntosh J, Marzo A, Fraser M, and Phillips C. EchoFlex: Hand Gesture Recognition using Ultrasound Imaging. In: *Proceedings of the 2017 CHI Conference on Human Factors in Computing Systems*. 2017:1923–34. DOI: [10.1145/3025453.3025807](https://doi.org/10.1145/3025453.3025807). URL: <https://doi.org/10.1145/3025453.3025807>.
10. Lykourinas A, Rottenberg X, Catthoor F, and Skodras A. Unsupervised Domain Adaptation for Inter-Session Re-Calibration of Ultrasound-Based HMIs. *Sensors* 2024;24.
11. Spacone G, Vostrikov S, Kartsch V, Benatti S, Benini L, and Cossettini A. Tracking of Wrist and Hand Kinematics with Ultra Low Power Wearable A-mode Ultrasound. *IEEE Transactions on Biomedical Circuits and Systems* 2024:1–13.
12. Lian Y, Lu Z, Huang X, et al. A Transfer Learning Strategy for Cross-Subject and Cross-Time Hand Gesture Recognition Based on A-Mode Ultrasound. *IEEE Sensors Journal* 2024;24:17183–92.
13. Kaifosh P, Reardon TR, and CTRL-labs at Reality Labs. A generic non-invasive neuromotor interface for human-computer interaction. *Nature* 2025.
14. Fournelle M, Grün T, Speicher D, et al. Portable Ultrasound Research System for Use in Automated Bladder Monitoring with Machine-Learning-Based Segmentation. *Sensors* 2021;21:6481.
15. Fournelle M, Grün T, Speicher D, Weber S, and Tretbar S. Portable low-cost 32-channel ultrasound research system. In: *2020 IEEE International Ultrasonics Symposium (IUS)*. 2020:1–3. DOI: [10.1109/IUS46767.2020.9251327](https://doi.org/10.1109/IUS46767.2020.9251327).
16. Zeng J, Zhou Y, Yang Y, Xu Z, Zhang H, and Liu H. Robustness of combined sEMG and ultrasound modalities against muscle fatigue in force estimation. In: *Lecture Notes in Computer Science*. Lecture notes in computer science. Cham: Springer International Publishing, 2021:213–21.
17. Zeng J, Zhou Y, Yang Y, Yan J, and Liu H. Fatigue-Sensitivity Comparison of sEMG and A-Mode Ultrasound based Hand Gesture Recognition. *IEEE Journal of Biomedical and Health Informatics* 2022;26:1718–25.
18. Zeng J, Sheng Y, Zhou Z, et al. Adaptive Learning Against Muscle Fatigue for A-Mode Ultrasound-Based Gesture Recognition. *IEEE Transactions on Instrumentation and Measurement* 2023;72:1–10.
19. Sgambato BG, Hasbani MH, Barsakcioglu DY, et al. High Performance Wearable Ultrasound as a Human-Machine Interface for Wrist and Hand Kinematic Tracking. *IEEE Transactions on Biomedical Engineering* 2024;71:484–93.

20. Iravantchi Y, Zhang Y, Bernitsas E, Goel M, and Harrison C. Interferi: Gesture Sensing using On-Body Acoustic Interferometry. In: Proceedings of the 2019 CHI Conference on Human Factors in Computing Systems. CHI '19. Glasgow, Scotland UK: Association for Computing Machinery, 2019:1–13. DOI: [10.1145/3290605.3300506](https://doi.org/10.1145/3290605.3300506). URL: <https://doi.org/10.1145/3290605.3300506>.
21. Farrell TR and Weir RF. The optimal controller delay for myoelectric prostheses. *IEEE Trans. Neural Syst. Rehabil. Eng.* 2007;15:111–8.

(Reply to Reviewers)

Virtual Reality Interactions via a User-Generic Ultrasound Human-Machine Interface for Wrist and Hand Tracking

Bruno Grandi Sgambato; Bálint K. Hodossy; Deren Yusuf Barsakcioglu; Xingchen Yang;
Anette Jakob; Marc Fournelle; Meng-Xing Tang; Dario Farina

We would like to express our appreciation and gratitude to the Reviewers for the time spent on reviewing the initial manuscript and the following version. We have further addressed the final comments by the reviewers below. A list of all changes is provided, with an explanation and justification for each modification and point-to-point answers to each comment. We believe these brief revisions addressed all final comments provided.

Reviewer 1

The authors provide helpful clarifications on the the stated objective of free movement of participants. Their intention was to contrast with highly restrictive posture constraints of the trunk and arm in prior work on ultrasound sensing of muscle activation, rather than aiming to support free movement across both body and the environment. This then allows for some integration of fixed optical tracking with the ultrasound system in order to demonstrate comparable performance to an all-optical approach. Essentially, the experimental set up is more focused on the reliability of the experimental design than the validity of the work against the proposed VR use case. The authors suggest elbow tracking may also be possible in future work from ultrasound data within the context of inverse kinematic constraint prediction. These arguments are convincing on their own merits, but they do suggest that, for fixed environments with relatively unconstrained movement, an ultrasound approach will struggle to demonstrate value in comparison to an optical system where multiple cameras can avoid typical issues such as occlusion, while fully unconstrained movement that could be enabled by an ultrasound system is beyond the capability reported in the paper.

The revisions which tighten up the precise scientific contributions of the work are positive. The authors have discriminated their work predicting multiple simultaneous targets from the existing literature which classifies larger discrete gesture sets. The key stand-out feature is the set of results which demonstrate independence and robustness in new participants. Again, this result doesn't place the approach in the same robustness bracket as optical VR approaches, but scoped according to ultrasound systems only is a clear advance.

The additions provided to enhance reproducibility of the work are appreciated, and through both the references and the additional characterisation, it will be possible to replicate the experimental setup described.

Overall, the paper now presents a stronger and clearer contribution to the field of ultrasonic gesture recognition. My remaining reservations relate to the proposed VR use case and the validity of the experimental studies as indicative of usage in this setting. It is quite clear that, at least for the capabilities demonstrated in this system, there will be no revolution in favour of ultrasound and away from optical approaches, and the approach still feels limited in production terms compared to existing robust hardware using EMG, visible or infrared signals. The work is provides a contribution to state of the art for ultrasonic gesture classification, but much less so for future VR approaches. For this reason the contribution may be too limited to warrant publication here.

Answer We thank the reviewer for further revising the manuscript and the changes made to address feedback. We were happy to hear the changes were positively received and have addressed the main concerns pointed in the initial review.

We agree with the feedback that the current system is not yet competitive with other EMG, or vision based methods in terms of production readiness. We however would posit that this strengthens the case for ultrasound. As mentioned, the independence and robustness on new participants was a stand-out finding from this work and its likely that many opportunities for improving it are still available as more research systems are developed and optimized for these applications. We hope that sharing these findings with a

the broader readership of the journal will attract more interest in the still niche topic of ultrasound-based gesture recognition and allow for further developments of the area.

Reviewer 2

The authors addressed all the previous comments. I only have one additional request for clarification related to angle normalization. The authors mention that normalization is performed on a per-session basis, normalizing the angles by the max value measured during that session. While this approach is clear for training, it's not clear how this works in testing on unseen data. Since the model works on normalized data, once the user starts to collect new data, what value is used for normalizing those input signals for the network? would it be the max value of the previous (known) session for that same user? A calibration session is first needed to evaluate what is the max range (for each DoF), as this is needed for the calibration?

Answer We thank the reviewer for following up on the changes we have made to address the feedback. We are happy to hear that the changes were satisfactory and have addressed the previous comments.

Regarding the normalization question, the reviewer is correct that normalization of the data was made on a per-session basis. For the training data this is straightforward as the minimum and maximum values for each session are easily extracted from the MOCAP data. It is indeed true that this would not be possible for new users/sessions were no MOCAP data was collected. However, there is no need to do so for the testing data. The training data ground truth is normalized so the model itself can be trained to output angles in a normalized range (what we called "normalized DoF activation values" or "pose values"). For system usage (testing sessions) no ground truth data is needed (as the model is not retrained/calibrated for each participant) and the ultrasound data can be feed to the model normally.

For the VR application explored this is completely appropriate as these normalized values can be readily mapped to arbitrary command ranges on the interface as needed. As the models are not theoretically restricted to provide predictions in the normalized range, it possible that predictions outside of it could be made. In our experiences this has not happened likely due to the loss function used (Squared L2 norm, mean squared error) heavily punishing outlier predictions during training. Anyhow, this could be easily dealt with by limiting predictions to be inside the normalized range. If the interface developed would be used in an different application (e.g., as a biomechanics research tool) where actual angles are required, then measurements of new users range of motion would indeed be required for the calculation of the absolute joint angles.

In the case of our offline evaluations, we made use of the range of motion measured from the testing sections to scale the predictions back from the normalized range to the true angle range. This was purely done so RMSE results could be shown in units of degree and be more easily analyzed and compared with other literature. For our online evaluations this procedure was not conducted and predictions from users were always shown in the normalized range.

Manuscript Changes: A reference to the loss function used was added to the Models and Model Training subsection. A comment better explaining that the range of motion reference data is not needed for using the system in the VR application was added to the Supplementary Discussion 9.

Reviewer 3

Answer We thank the reviewer for its efforts in thoroughly reviewing the manuscript. We hope to have answered your final questions in the answers above.